# Multi-night cortico-basal recordings reveal mechanisms of NREM slow-wave suppression and spontaneous awakenings in Parkinson's disease

Md Fahim Anjum [1] ✉, Clay Smyth[1], Rafael Zuzuárregui[1,2], Derk Jan Dijk [3,4], Philip A. Starr [1], Timothy Denison[5] & Simon Little [1]

Sleep disturbance is a prevalent and disabling comorbidity in Parkinson's disease (PD). We performed multi-night (n = 57) at-home intracranial recordings from electrocorticography and subcortical electrodes using sensing-enabled Deep Brain Stimulation (DBS), paired with portable polysomnography in four PD participants and one with cervical dystonia (clinical trial: NCT03582891). Cortico-basal activity in delta increased and in beta decreased during NREM (N2 + N3) versus wakefulness in PD. DBS caused further elevation in cortical delta and decrease in alpha and low-beta compared to DBS OFF state. Our primary outcome demonstrated an inverse interaction between subcortical beta and cortical slow-wave during NREM. Our secondary outcome revealed subcortical beta increases prior to spontaneous awakenings in PD. We classified NREM vs. wakefulness with high accuracy in both traditional (30 s: 92.6 ± 1.7%) and rapid (5 s: 88.3 ± 2.1%) data epochs of intracranial signals. Our findings elucidate sleep neurophysiology and impacts of DBS on sleep in PD informing adaptive DBS for sleep dysfunction.

Sleep disruption is one of the most prevalent non-motor symptoms of Parkinson's disease (PD) with up to 90% of PD patients experiencing sleep dysfunction[1] and 60% having multiple sleep disturbance symptoms[1,2]. Sleep dysfunction in PD has a negative impact on daytime mood, cognition, fatigue, and other co-morbidities[3–7], with non-motor and sleep-related symptoms being a greater determinant of quality of life than classical motor symptoms[8–10]. Changes in sleep patterns often predate classical neurological symptoms in PD and overnight slow-wave dysfunction correlates with rates of disease progression and severity[11,12]. Therefore, understanding the neurophysiology of sleep disturbances in PD may potentially result in new principled therapies directed toward better sleep quality, mitigation of daytime symptoms,

improved patients' quality of life and the development of therapeutic targets for disease progression modification.

Sleep architecture in humans is broadly defined by physiologically distinct stages of rapid eye movement (REM) and non-REM (NREM) sleep. NREM sleep is further characterized by rhythmic low-frequency electroencephalography (EEG) activity in the delta (0–4 Hz) and theta (4–8 Hz) ranges, increased parasympathetic activity, and limited dreaming. There are currently three formally defined sub-stages of NREM: N1 (light sleep), N2 (appearance of K complexes and sleep spindles) and N3 (characterized by slow delta waves)[13]. Sleep dysfunction in PD manifests as dream enactment behavior, fragmented sleep and disrupted sleep patterns, including notable reductions in

[1]Movement Disorders and Neuromodulation Centre, University California San Francisco, San Francisco, CA, USA. [2]Parkinson's Disease Research Education and Clinical Center, San Francisco Veteran's Affairs Medical Center, San Francisco, CA, USA. [3]Surrey Sleep Research Centre, University of Surrey, Guildford, UK. [4]UK Dementia Research Institute, Care Research and Technology Centre at Imperial College, London and The University of Surrey, Guildford, UK. [5]MRC Brain Network Dynamics Unit, University of Oxford, Oxford, UK. ✉ e-mail: fahim.anjum@ucsf.edu

both REM and NREM sleep[10]. In particular, reductions in NREM slow wave activity in the delta range (<4 Hz) are associated with worsening of daytime motor symptoms and accelerated disease progression in PD[11,14,15].

During wakefulness, beta oscillations (13–31 Hz) are the hallmark oscillatory signature of PD and correlate with daytime motor symptoms[16]. Recent studies with non-human primates (NHPs) during sleep have shown that subcortical beta activity is also associated with a decrease in cortical delta activity and suggested a role for subcortical beta in spontaneous awakenings in PD[17]. Indeed, the presence of sub-cortical beta oscillations in subthalamic nucleus (STN) and globus pallidus (GPi) has been detected during sleep in PD patients[18–23]. However, these studies have been single-night studies, recorded in externalized patients post-operatively in a laboratory. To date, human studies have not yet investigated mechanistic interactions between subcortical beta and cortical sleep physiology (inc. slow waves), spontaneous awakenings within individuals with PD nor investigated DBS stimulation effects on sleep physiology with high temporal precision. In summary, past studies have been limited to single-night, across-participant analyses that have not yet determined if subcortical beta is a reliable biomarker of awakening within individuals, or in the presence of deep brain stimulation (DBS).

Understanding the real-world contribution of the cortico-basal ganglia circuit to sleep dysfunction in PD and its interaction with DBS has been limited by an inability to chronically record the intracranial activities overnight, at high resolution. This challenge has been mitigated by the advent of a new generation of sensing-enabled DBS devices that can stream neural data remotely from participants' own homes[24]. A better understanding of cortico-basal activities during sleep has the potential to reveal underlying mechanisms of sleep dysfunction in PD and could contribute to improved sleep therapies including sleep-targeted adaptive deep brain stimulation (aDBS) for neurological disorders[25].

In this study, we recruited four participants diagnosed with PD and one comparison participant with cervical dystonia, all with chronically implanted intracranial electrodes capable of sensing sensorimotor cortical and basal ganglia (STN and GPi) field potentials (FPs). We conducted large data collection, within-participant, multi-night, at-home, intracranial cortical and subcortical recordings paired with portable polysomnography over multiple nights (n = 57) in the presence and absence of DBS stimulation. We demonstrate significant negative interactions between subcortical beta oscillations and cortical slow wave activity in the delta band during N2/N3 NREM, an effect modulated by DBS, and also show that subcortical beta significantly increases prior to spontaneous awakenings at high-temporal resolution (5 s time window). Finally, we demonstrate successful classification of N2/N3 NREM vs. wakefulness in both classical (30 s) and sub-classical (5 s) time windows using constrained machine-learning approaches using bandpower features from intracranial neural recordings–toward the development of sleep-specific adaptive DBS therapies for neurological disorders. Our findings on the mechanisms of cortical-subcortical interactions during sleep provide a foundation for the development of adaptive DBS approaches for restoring physiological sleep patterns in people with PD.

## Results
Four participants (Table 1) with PD (x2 with bilateral STN + sensorimotor cortical ECoG and x2 with bilateral GPi electrodes + sensorimotor cortical ECoG) and one participant with cervical dystonia (bilateral GPi electrodes + sensorimotor cortical ECoG), successfully initiated recordings from intracranial cortico-basal and external portable polysomnography (Dreem2[26,27]) over a total of 57 nights (53 ON and 4 OFF stimulation nights), remotely in their own homes. Intracranial and extracranial recordings were synchronized by resampling the extracranial (Dreem2 headband) data and determining the delays

(lag) between the intracranial and extracranial signals by the application of cross-correlation to the accelerometry data from both sources (Fig. 1E and Supplementary Fig. 3). As a secondary validation, included in the protocol were pre-planned synchronized perturbations (x5 taps) of the accelerometers on both the RC + S and the wearable PSG (this secondary validation was performed independently and bilaterally for the two hemispheres) and the final synchronization outcomes were manually inspected for each night. Large artifactual spikes in the subcortical intracranial data were detected by first smoothing the absolute squared data with Gaussian kernel and then finding time periods that exceeded a threshold (determined for each night; see "Methods" for details). These were removed along with the corresponding cortical data (Supplementary Fig. 3). Finally, the ECG artifacts in the subcortical data were removed using an optimized combination of two ECG data removal algorithms[28,29] (Supplementary Fig. 3; see "Methods" for details) resulting in interpretable cortical and subcortical recordings, even in the presence of DBS.

A total 407 h of sleep were recorded across all participants (Supplementary Table 1 and Supplementary Fig. 1). Polysomnography (PSG) data were collected from the Dreem2 PSG headband which provided automated sleep scoring for 30 s epochs according to standard sleep staging (Wake, NREM: N3, N2, N1 and REM) via automated EEG-based sleep staging algorithm which has previously been validated on healthy participants (Fig. 1C)[26,27]. As N1 generally is difficult to detect and physiologically distinct, we focused our analysis on N2 and N3 stages for NREM sleep (denoted as N2/N3 NREM). Finally, N2/N3 NREM to wakefulness captured by polysomnography were manually re-scored by a board certified sleep physician to corroborate the automated scoring and obtain more precise awakening estimates. PD participants slept on average 7.23 ± 0.19 h per night during the multi-night ON stimulation recording phase (n = 44; total duration in minutes per night: N1 = 34.11 ± 1.38; N2 = 164.54 ± 8.07; N3 = 92.41 ± 10.67; REM = 94.89 ± 6.49; Wake after sleep = 46.59 ± 4.53). In a separate two-night, consecutive ON versus OFF DBS comparison, all four PD participants showed an increase in time of N3 and REM sleep during ON stimulation compared to the OFF stimulation nights (Supplementary Table 1 and Supplementary Fig. 2). Power spectral density plots from intracranial electrodes (Fig. 1G and Supplementary Fig. 4) demonstrated expected classical changes in canonical frequency bands in NREM and REM sleep stages, supporting appropriate dissociation of different sleep stages using portable PSG device sleep staging.

### Spectral power changes in NREM
We investigated overnight spectral changes in intracranial activities, specifically investigating the hypothesis that there is a negative interaction between cortico-basal delta and beta during N2/N3 NREM[30]. Power spectrum analyses and Linear Mixed Effect (LME) models for average overnight band powers with a fixed effect for sleep stage (N2/N3 NREM vs. Wake; accounting for multiple nights within participants) and a random effect for participants (n = 5) showed a decrease in average beta power (13–31 Hz; cortex: $\beta = -0.41$, 95% CI = [−0.44, −0.38], p value = 8.4e−45; subcortex: $\beta = -0.23$, 95% CI = [−0.27, −0.2], p value = 3.1e−27) and increase in delta power (1–4 Hz; cortex: $\beta = 0.43$, 95% CI = [0.39, 0.48], p value = 3.2e−37; subcortex: $\beta = 0.1$, 95% CI = [0.08, 0.13], p value = 5.5e−14; n = 106; CI = confidence interval) both in cortical and subcortical regions in N2/N3 NREM sleep compared to wakefulness (Fig. 2A, B; multi-night ON stimulation). These spectral changes in N2/N3 NREM compared to wake were also observed during the single night of OFF DBS sleep recordings in both cortical and subcortical regions of all four PD participants (Fig. 2C).

A direct comparison of PD (n = 4) vs. Dystonia (n = 1) revealed that subcortical beta power was lower in the dystonia than all four of the PD participants during N2/N3 NREM sleep (LME model for PD vs. Dystonia

**Table 1 | Participant demographics, clinical characteristics and stimulation settings**

| Participant ID | PD2 | PD3 | PD7 | PD9 | Dystonia |
|---|---|---|---|---|---|
| Demography | | | | | |
| Age | 58 | 66 | 40 | 48 | 65 |
| Gender | M | M | M | M | M |
| Diagnosis | PD | PD | PD | PD | Dystonia |
| Dx | 11 | 13 | 9 | 13 | 30 |
| DBS stimulation settings | | | | | |
| Stim target | STN | GPi | STN | GPi | GPi |
| Pulse width (us) | 60 | 60 | 60/90 | 90 | 60 |
| Stim amp. (mA) | L: 2.4 R: 3.1 | L: 3.7 R: 2.8 | L: 1.7–3.4 R: 1.7–3.4 | L: 3–3.7 R: 3–3.7 | L: 4.5 R: 3.5 |
| Stim freq. (Hz) | 130.2 | 178.6 | 130.2 | 150.6 | 130.2 |
| Stim contact | L: C+2− R: C+1− | L: C+1− R: C+1− | L: C+2− R: C+2− | L: C+2− R: C+2− | L: C+1− R: C+2− |
| Symptoms and clinical characteristics | | | | | |
| Medication details | A-HCL 100 mg (3 times daily) C-Ldopa 25–100 mg IR (5 times daily) | C-Ldopa 25–100 mg CR (1–2 tabs at bedtime) and 25–100 mg IR (3 times daily) | C-Ldopa 25–100 mg (1 time daily) Rasagiline (Azilect) 1 mg (1 time daily) | Rytary 195 mg (3 times daily) | – |
| UPDRS-III (OFF) | 49 | 66 | 41 | 39 | – |
| UPDRS-III (ON) | 5 | 24 | 14 | 16 | – |
| UPDRS 1.7 | No sleep symptoms | Slight sleep symptoms | Slight sleep symptoms | Mild sleep symptoms | – |
| UPDRS 1.8 | No daytime sleepiness | Mild daytime sleepiness | Moderate daytime sleepiness | Mild daytime sleepiness | – |
| Sleep diagnosis | No sleep conditions | Nocturia, RBD | Daytime sleepiness | OSA, Insomnia | Restless Leg Syndrome |
| Neuropsych report (pre-op) | No reported sleep disorder or conditions | Mild sleep difficulties, with nocturia and RBD | Day time sleepiness (strongest 4–5 pm), usually sleeps late and sleeps very little overnight | Had long-term difficulties sleeping before PD.Occasionally couldn't fall asleep at night. | Good sleep. No movements /dystonia at night. Restless Leg Syndrome at night. |

*UPDRS* Unified Parkinson's Disease Rating Scale, *Dx* disease duration (years), *RBD* REM sleep behavior disorder, *OSA* obstructive sleep apnea, *PD* Parkinson's disease, *R* right, *L* left, *C-Ldopa* Carbidopa-Levodopa (Sinemet), *A-HCL* Amantadine HCL (Symmetrel), *Stim* DBS stimulation. UPDRS scores were pre-operative. None of the participants suffered from Dementia.

fixed effect: $\beta = 0.19$; 95% CI = [0.11, 0.28]; $p$ value = 2.3e−5; $n = 53$). A similar comparison of cortical beta did not show any statistically significant changes at the group level during N2/N3 NREM sleep ($p$ value = 0.34). However, during wakefulness both cortical and subcortical beta was higher in PD compared to dystonia (LME model for PD vs. Dystonia fixed effect; cortical beta: $\beta = 0.14$; 95% CI = [0.015, 0.26]; $p$ value = 0.028; subcortical beta: $\beta = 0.36$; 95% CI = [0.31, 0.41]; $p$ value = 2.4e−19). Further, band power changes between N2/N3 NREM sleep and wakefulness conditions in the dystonia participant were smaller compared to the PD participants. Indeed, LME models demonstrated statistically significant fixed effects of disease state (PD vs. Dystonia) on the changes of band power between N2/N3 NREM and wake stage in cortex (delta: $\beta = 0.24$, 95% CI = [0.15, 0.34], $p$ value = 4.3e−6; beta: $\beta = -0.21$; 95% CI = [−0.28, −0.14], $p$ value = 1.7e−7) and subcortex (delta: $\beta = 0.08$, 95% CI = [0.017, 0.14], $p$ value = 0.01; beta: $\beta = -0.17$, 95% CI = [−0.24, −0.09], $p$ value = 3.5e−5), supporting more pronounced N2/N3 NREM vs. wake changes in PD vs. the Dystonia participant.

We also investigated how these activities alter with DBS during N2/N3 NREM and compared power spectrums between ON and OFF stimulation conditions in our PD cohort ($n = 4$; Table 1). Spectral power comparisons revealed a relative further increase in delta and further decrease in alpha and sigma activities in cortical region during N2/N3 NREM sleep in the ON vs. OFF DBS conditions (Fig. 2D). Indeed, LME models with participants as random effects revealed that stimulation (ON DBS) resulted in a further increased cortical delta (1–4 Hz; $\beta = 0.027$, 95% CI = [0.0003, 0.05], $p$ value = 0.03) and decreased cortical alpha (8–13 Hz; $\beta = -0.031$, 95% CI = [−0.06, −0.004], $p$ value = 0.03) as well as sigma (13–15 Hz; $\beta = -0.03$, 95% CI = [−0.049, −0.01],

$p$ value = 0.01) in N2/N3 NREM for the ON vs. OFF condition. Location of the DBS leads in the subcortical structure (GPi/STN) did not show any statistically significant effect in the changes of cortical delta ($p$ value = 0.06) and alpha ($p$ value = 0.4) but had a statistically fixed effect only in decreased sigma power (13–15 Hz; $\beta = -0.56$, 95% CI= [−1.02, −0.1], $p$ value = 0.026) in GPi compared to STN. No significant changes in subcortical activities during N2/N3 NREM were observed in ON vs. OFF power spectrum comparisons, however, changes in subcortical baseline power levels in the ON vs. OFF DBS state and lower SNR (signal-to-noise ratio) in data from GPi of PD3 (ON state) may have obscured any underlying changes. Overall, these data reveal that DBS results in relatively higher cortical delta activity and reduced alpha and low-beta activities in N2/N3 NREM sleep.

Finally, we investigated the impact of dopaminergic medications by dividing each night into four quadrants and measuring effects in the neurophysiology at the end of the night (when the dopaminergic medications had partially worn off) compared to the beginning of the night. We utilized data from 1st and 4th quadrant of all 4 PD participants during ON stimulation and implemented a linear mixed effect (LME) model of cortical delta with subcortical beta and time quadrant (1st or 4th) as fixed effects. The LME model showed a statistically significant fixed effect of quadrant (beginning/end of the night) on the cortical delta ($\beta = -0.01$, 95% CI = [−0.02, −0.006], $p$ value = 1.7e−5, $n = 99,155$) while subcortical beta showed a statistically significant negative effect on cortical delta as expected while accounting for both quadrants ($\beta = -0.44$, 95% CI = [−0.49, −0.4], $p$ value = 3.2e−87). The negative effect of subcortical beta was significantly stronger during the 4th quadrant (effect of 1st/4th quadrant on the interaction: $\beta = -0.006$, 95% CI = [−0.01, −0.002], $p$ value = 0.003).

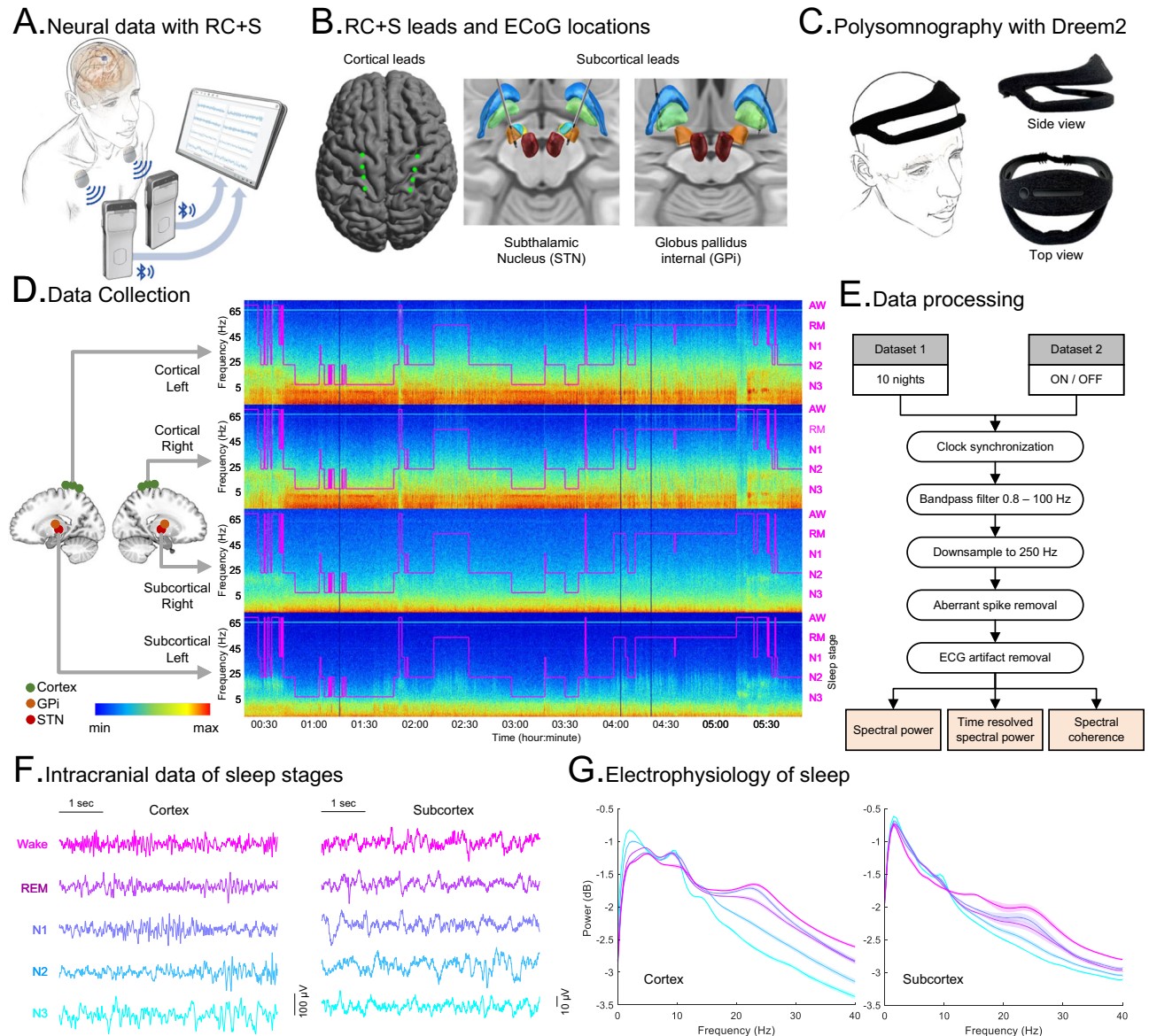

**Fig. 1 | Methodology, data collection and analysis procedures. A** Schematic of the RC + S system setup for recording intracranial cortical Field Potentials (FP) in participants. **B** Illustrations of the placement of RC + S sensing depth electrodes in subcortex (middle and right) for both Subthalamic Nucleus (STN) and Globus pallidus internal (GPi) and cortical ECoG locations (left). Example image from PD2 and PD3 participants. **C** Setup of the Dreem2, portable headband for recording in-home overnight polysomnography. **D** Illustration of a single night of sleep recording in a PD participant (DBS ON) with polysomnography (purple) showing sleep stages (right *y*-axis; AW: awake; RM: REM; [N1, N2, N3]: NREM) and simultaneous cortical (top 2 panels) and subcortical (bottom 2 panels) spectrogram of FPs from both hemispheres showing multi-frequency changes across sleep stages where the *x*-axis is time and *y*-axis (left) is frequency (Hz). FP was recorded bilaterally from cortical and subcortical regions. **E** Flowchart of data analysis and preprocessing procedures for multi-night sleep dataset of all participants (*n* = 5; ~10 nights per participant) and ON/OFF dataset (2 nights per participant) of PD participants (*n* = 4). **F** Representative traces of the RC + S FP time series (5 s epochs) in all sleep stages from cortex (left column) and subcortex (right column; STN). Columns share scale bars and rows share color legends (Wake, REM, N1, N2, and N3). Data from one PD participant (PD2) with ON stimulation from the left hemisphere. **G** Comparisons of spectral powers of intracranial FPs among sleep stages in cortex (left) and subcortex (right) for a single participant across multiple nights (*n* = 12; PD2; DBS ON; 5 s epochs; averaged across each night; data pooled from both hemispheres; shares color legend with **F**). Data are presented as mean ± SEM. Spectral comparisons for all participants are provided in Supplementary Fig. 4. Source data are provided as a Source Data file.

## Changes in functional connectivity in NREM

We next explored NREM-related changes in the functional connectivity between sensorimotor cortical and subcortical regions to investigate sleep-related changes in cortico-basal ganglia circuitry in PD. For this, we compared the spectral coherence in cortical and subcortical activities between N2/N3 NREM sleep and wakefulness. In all participants, LME models investigating spectral coherence with a fixed effect of sleep stage (N2/N3 NREM vs. Wake) revealed that the total

difference in spectral coherence in beta decreases ($\beta = -0.19$; 95% CI = [−0.23, −0.14], *p* value = 5.4e−13) while in delta increases ($\beta = 0.05$, 95% CI = [0.04, 0.06], *p* value = 7e−13; *n* = 105) during N2/N3 NREM sleep compared to wake, ON stimulation (Fig. 2E, F). A decrease in beta coherence and increase in delta coherence during N2/N3 NREM were also observed in the PD participants during their single night recordings OFF stimulation (Fig. 2G). PD vs. Dystonia comparison also showed that cortico-basal delta/beta coherence changes in N2/N3

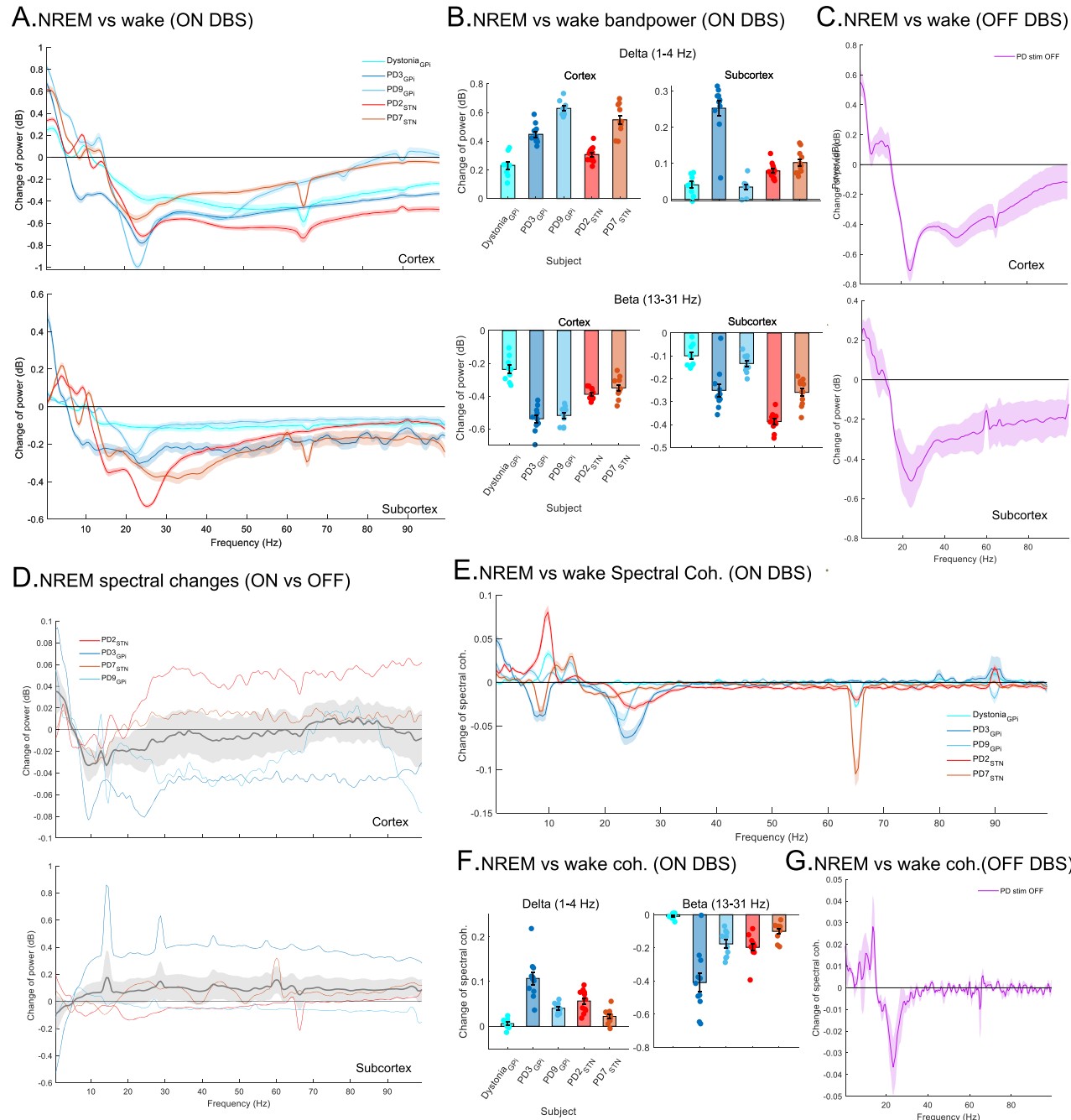

**Fig. 2 | Dynamic changes in power spectra and functional connectivity between cortical and subcortical regions during N2/N3 NREM sleep. A** Power spectrum changes during N2/N3 NREM with wake as baseline for all participants ($n = 5$) during ON stimulation in cortical (top) and subcortical (bottom) areas. $y$-axis shows the difference in power spectra (mean ± SEM; dB) between N2/N3 NREM and wakefulness. **B** Power in delta increases (top) while beta decreases (bottom) during N2/N3 NREM compared to wakefulness during ON stimulation in cortical (left) and subcortical (right) areas. Bar plots show the difference in spectral power (mean ± SEM; each data point shows average difference in dB across one night). **C** During OFF stimulation, delta power increases while beta decreases in N2/N3 NREM compared to the wakefulness in PD participants ($n = 4$) in cortical (top) and subcortical (bottom) areas (difference in power spectra in dB; mean ± SEM). **D** Difference in cortical spectral power between ON and OFF stimulation (ON power-OFF power; each colored line for one participant; mean ± SEM in gray) in 4 PD participants in N2/N3 NREM (top), showing increased delta and decreased alpha and sigma

activities (8–15 Hz) while ON stimulation. The spectral power in subcortical regions didn't show any statistically significant difference (bottom). **E** Changes in cortical-subcortical spectral coherence (mean ± SEM) during N2/N3 NREM with wakefulness as baseline for all participants ($n = 5$) during ON stimulation. $y$-axis shows the difference in spectral coherence between N2/N3 NREM and wakefulness. **F** Total difference in spectral coherence in delta (left) and beta (right) during N2/N3 NREM compared to wake during ON stimulation. Barplots show difference in spectral coherence (mean ± SEM; each point shows average difference in spectral coherence across one night). **G** During OFF stimulation, delta coherence increases while beta coherence decreases in N2/N3 NREM compared to the wakefulness in PD participants ($n = 4$; mean ± SEM). Data from both hemispheres were pooled for all panels. Baseline is shown as horizontal line at 0 for **A**, **C**, **D**, **E**, and **G**. For **B** and **F**: $n = 12$ (PD2); $n = 11$ (PD3); $n = 11$ (PD7); $n = 10$ (PD9); $n = 9$ (Dystonia). Source data are provided as a Source Data file.

NREM versus wakefulness were smaller in the dystonia participant compared to the PD participants (LME model with PD/Dystonia condition as a fixed effect; delta coherence: $\beta = 0.05$, 95% CI = [0.02, 0.08], $p$ value = 0.0005; beta coherence: $\beta = -0.21$, 95% CI = [−0.32, −0.11], $p$ value = 0.0001).

In our ON vs. OFF DBS analysis (PD participants only; $n = 4$; Table 1), we also noted a statistically significant further decrease in cortico-basal sigma (13–15 Hz) coherence during ON stimulation compared to OFF (LME model with ON/OFF condition as fixed and participants as random effects; $\beta = -0.014$, 95% CI = [−0.021, −0.006], $p$ value = 0.005) during N2/N3 NREM with a statistically fixed effect of the location of DBS leads (STN/GPi: $\beta = 0.08$, 95% CI = [0.002, 0.16], $p$ value = 0.047). Collectively, these data demonstrate that functional connectivity between cortical and subcortical structure is modulated during N2/N3 NREM sleep compared to wakefulness. Specifically, there is a decrease in beta coherence and an increase in delta coherence in PD during N2/N3 NREM in both ON and OFF stimulation conditions with wakefulness as the baseline, effects that are enhanced in the DBS ON condition.

## Interaction between cortical delta and subcortical beta activity

Spectral power and functional connectivity analyses above revealed opposing changes in delta and beta activities in N2/N3 NREM sleep vs. wakefulness. To further examine for a direct relationship between these two rhythms, we investigated the interactions between cortical delta and subcortical beta activities specifically *within* N2/N3 NREM on shorter, within-sleep stage, time scales (5 s). Here, we observed an inverse relationship between cortical delta power and subcortical beta power during N2/N3 NREM sleep (Fig. 3A). To quantify this relationship, we first used a within-participant analysis which revealed a negative correlation between subcortical beta and cortical delta power (5 s epochs) in all PD participants during N2/N3 NREM in both ON and OFF stimulation conditions (Fig. 3B, C). LME models for each PD participant individually showed a negative fixed effect of subcortical beta on cortical delta during N2/N3 NREM sleep in ON stimulation, across all nights (PD3: $\beta = -0.43$, $p$ value = 1.24e−19; PD9: $\beta = -0.59$, $p$ value = 7.1e−39; PD2: $\beta = -0.47$, $p$ value = 1.49e−54; PD7: $\beta = -0.55$, $p$ value = 1.48e−13). Furthermore, LME modeling using band powers of N2/N3 NREM epochs from all participants (Cervical dystonia and PD participants; accounting for the dependency between left and right hemispheres and multiple nights within participants; $n = 241,643$) showed an overall negative fixed effect of subcortical beta power on cortical delta power ($\beta = -0.36$, 95% CI: [−0.42, −0.3], $p$ value = 2.5e−30) during N2/N3 NREM sleep, ON stimulation. Additionally, the LME model revealed a fixed effect of PD vs. Dystonia state ($\beta = 0.16$, 95% CI: [0.1, 0.22], $p$ value = 1.7e−7), demonstrating that this effect was greater in the PD participants than our dystonia comparison participant. A negative fixed effect of subcortical beta power on cortical delta power was also obtained through an LME model in PD participants during N2/N3 NREM in the OFF stimulation condition ($\beta = -0.4$, 95% CI: [−0.49, −0.31], $p$ value = 2.5e−17; $n = 18,226$). These results demonstrate that there is an inverse relationship between subcortical beta and cortical delta power within N2/N3 NREM sleep in PD both during ON and OFF stimulation conditions and that this effect is significantly stronger than in our comparison dystonia participant.

Next, we utilized cross-correlation analyses to determine whether subcortical beta was leading or lagging cortical delta changes. We observed that the subcortical beta increase was leading the cortical delta decrease in 3 out of the 4 PD participants during N2/N3 NREM sleep (Fig. 3D; average lag over multiple nights ON DBS; PD2: 5.4 s, $n = 12$; PD3: −9.1 s, $n = 11$; PD7: −6.4 s, $n = 11$; PD9: −4.5 s, $n = 10$). Finally, as a control analysis to rule out a prosaic inverse relationship between cortico-basal circuit delta and beta, simply reflecting the depth of NREM sleep, we also measured the interaction between cortical delta and cortical beta power from the same region. If the inverse

relationship between cortico-basal delta and beta was simply a function of sleep stage depth, we would also expect a strong inverse relationship between cortical delta and cortical beta. Unlike correlations between subcortical beta and cortical delta, which were negative for all PD participants, cortical delta and beta showed a weaker negative correlation in 3 PD participants and a positive correlation in one PD participant (Fig. 3E) as well as in the Dystonia participant during N2/N3 NREM (ON stimulation). LME model for 3 out of 4 PD participants showed negative fixed effects of cortical beta on cortical delta during N2/N3 NREM sleep in ON stimulation (PD3: $\beta = -0.67$, $p$ value = 1.72e−54; PD9: $\beta = -0.36$, $p$ value = 1.9e−9; PD2: $\beta = 0.23$, $p$ value = 9.4e−11; PD7: $\beta = -0.43$, $p$ value = 2.3e−20). Across all participants, LME analysis did show a weaker overall negative fixed effect of cortical beta power on cortical delta power ($\beta = -0.23$, 95% CI: [−0.33, −0.12], $p$ value = 3.4e−5; $n = 241,643$) during N2/N3 NREM sleep with a fixed group effect of PD/Dystonia state ($\beta = -0.04$, 95% CI: [−0.07, −0.013], $p$ value = 0.004). Additionally, in direct model comparison, the LME model for cortical delta with a fixed effect of subcortical beta showed a statistically significant improvement over the model of cortical delta with a fixed effect of cortical beta (simulated likelihood ratio test with 100 replications; $p$ value = 0.01). This demonstrates that subcortical beta had a stronger effect on cortical delta compared to the relationship between cortical beta activity and cortical delta activity supporting that this subcortical beta−cortical delta effect is greater than any effect of sleep stage depth.

## Changes in spectral power before spontaneous awakenings

To better understand neural activities at a finer time resolution and investigate the dynamics of intracranial neurophysiology that lead to awakenings, we analyzed the change in spectral powers in delta and beta during NREM to wake transitions. There were a total of $25.20 \pm 1.9$ awakenings per night from all sleep stages (including N1 to wake transitions) with an average duration of $2.07 \pm 0.23$ min for PD participants during ON stimulation. For N2/N3 NREM specifically, there was a total of $12 \pm 1.1$ N2/N3 NREM to wake transitions per participant, per night, with each spontaneous awakening averaging $3.70 \pm 0.52$ min for PD participants.

In our time-resolved analysis of N2/N3 NREM to wakefulness transitions for all ($n = 5$) participants (multi-night ON stimulation dataset), the subcortical beta power demonstrated a rise before awakenings which was further sustained after awakening (Fig. 4A; immediate pre-wake N2/N3 NREM: $t = -7.5$ s, $\beta = 0.35$, 95% CI: [0.16, 0.54], $p$ value = 0.0003, $n = 1022$; early post-wake: $t = +12.5$ s, $\beta = 1.6$, 95% CI: [0.53, 2.7], $p$ value = 0.003) and post-awakening power was higher than pre-awakening N2/N3 NREM ($\beta = 1.6$ vs. 0.35). Disease state (PD/dystonia) showed a statistically significant fixed effect on the rise of subcortical beta power before awakenings ($\beta = -0.87$, 95% CI: [−1.5, −0.24], $p$ value = 0.007) which was not significant after awakenings (early post-wake: $p$ value = 0.21) suggesting that the rise of subcortical beta power in the immediate pre-awakening N2/N3 NREM is relatively PD specific. Conversely, cortical beta power did not show any statistically significant changes in the immediate pre-awakening N2/N3 NREM or in early post-wake periods (immediate pre-wake N2/N3 NREM: $t = -7.5$ s; $p$ value = 0.07; early post-wake: $t = +12.5$ s, $p$ value = 0.45; Fig. 4B).

During N2/N3 NREM, we found that cortical delta power gradually increases as sleep deepens (Fig. 4D) in all participants. The cortical delta power in the early post-awakening ($t = +12.5$ s) periods was lower compared to the delta power found in deep N2/N3 NREM stage, (Fig. 4D; early post-wake: $\beta = -5$, 95% CI: [−6, −4], $p$ value = 2.4e−20, $n = 1022$) and no fixed effects of PD/dystonia condition ($p$ value = 0.51). Our analyses didn't show any statistically significant changes in cortical delta in the immediate pre-awakening N2/N3 NREM compared to the deep N2/N3 NREM ($t = -7.5$ s; $p$ value = 0.54; Fig. 4D). The subcortical delta showed statistically significant post-awakening changes compared to deep N2/N3 NREM (early post-wake in subcortical delta:

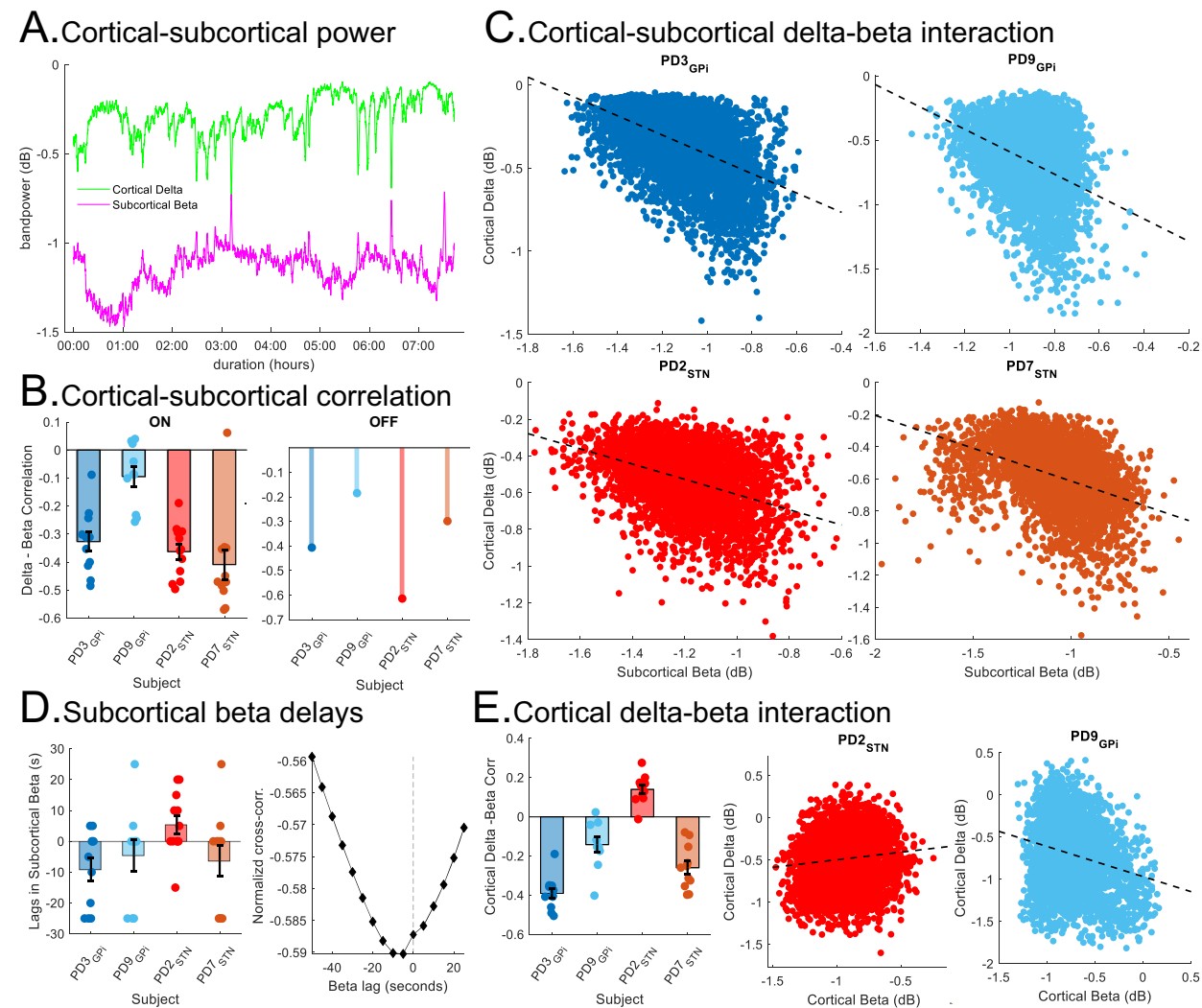

**Fig. 3 | Inverse relationship between subcortical beta and cortical delta activities during N2/N3 NREM sleep. A** Example of subcortical beta (purple) and cortical delta (green) power (PD3; single night; ON stimulation; smoothed with 20-point Gaussian kernel) depicting the inverse relationship. **B** Average Spearman's rho correlation between subcortical beta and cortical delta power for 4 PD participants in ON (left; mean ± SEM; each point shows overnight correlation) and OFF (right; stem plots; $n = 4$; single night per participant) stimulation. **C** Scatter plots depicting the correlation between subcortical beta (STN: brown, red; GPi: blue, light blue) and cortical delta power in 4 PD participants (ON stimulation; 5 s epochs; each plot is single night data pooled from both hemispheres). LME models constructed for cortical delta with subcortical beta as fixed and hemisphere as random effect (PD3: $\beta = -0.58$, $p$ value = 0; PD9: $\beta = -0.87$, $p$ value = 1.2e−144; PD2: $\beta = -0.41$, $p$ value = 1.8e−129; PD7: $\beta = -0.41$, $p$ value = 1.3e−139). **D** Normalized cross-correlation between subcortical beta and cortical delta power (mean ± SEM) showing the subcortical beta preceding cortical delta in PD participants (ON

stimulation). The bar plot (left; each point shows overnight lag) shows lags in subcortical beta compared to cortical delta. Example of cross-correlation showing the lag in subcortical beta with normalized cross-correlation vs. lag time (s) for subcortical beta with cortical delta as reference (right; PD2; single night; ON stimulation; dashed vertical line is zero-lag). **E** Interactions between cortical delta and beta. The bar plot (left) shows average Spearman's rho correlation between cortical delta and beta power (mean ± SEM; 4 PD participants across multiple nights; each point shows overnight correlation; ON stimulation). The scatter plots (middle and right) show cortical delta and beta power (ON stimulation; 5 s epoch; single nights for PD2 and PD9). LME models were similar to **C** (PD2: $\beta = 0.18$, $p$ value = 4.7e−12; PD9: $\beta = -0.36$, $p$ value = 1.7e−60). For barplots in **B** (ON stimulation), **D** and **E**: data pooled from both hemispheres with $n = 12$(PD2), $n = 11$(PD3), $n = 11$(PD7), and $n = 10$(PD9). Data from all panels are from N2/N3 NREM. All $p$ values were two-sided. Source data are provided as a Source Data file.

$t = +12.5$ s; $p$ value = 0.034; Fig. 4C) with no fixed effects of PD/dystonia condition ($p$ value = 0.42) and did not demonstrate pre-awakening N2/N3 NREM changes that were statistically significant (immediate pre-wake N2/N3 NREM in subcortical delta: $t = -7.5$ s; $p$ value = 0.36; Fig. 4C). These data support that changes in cortical and subcortical delta during N2/N3 NREM to wake after sleep transitions are not PD specific, but rather a general feature of changes in neurophysiology in NREM sleep vs. wakefulness.

### Classifying NREM vs. wakefulness and spontaneous awakenings

We next investigated classification performance using machine-learning (ML) models on intracranial neurophysiology to distinguish

N2/N3 NREM and wakefulness. To accomplish this, we utilized data from the sensorimotor cortical and subcortical regions of 4 PD participants, during the multi-night ON DBS dataset and trained participant-specific support vector machine (SVM) classifiers with six bandpowers features including: delta (0–4 Hz), theta (4–8 Hz), alpha (8–13 Hz), sigma (13–15 Hz), high beta 15–31 Hz) and low gamma (31–50 Hz).

First, we trained a participant-specific SVM model for each PD participant ($n = 4$) in classical 30 s time windows to classify all N2/N3 NREM vs. wake epochs using first cortical and then subcortical features (Fig. 5A). Our results showed strong classification performance both using sensorimotor cortex and subcortical regions (Cortex: 93.6 ± 1.4%

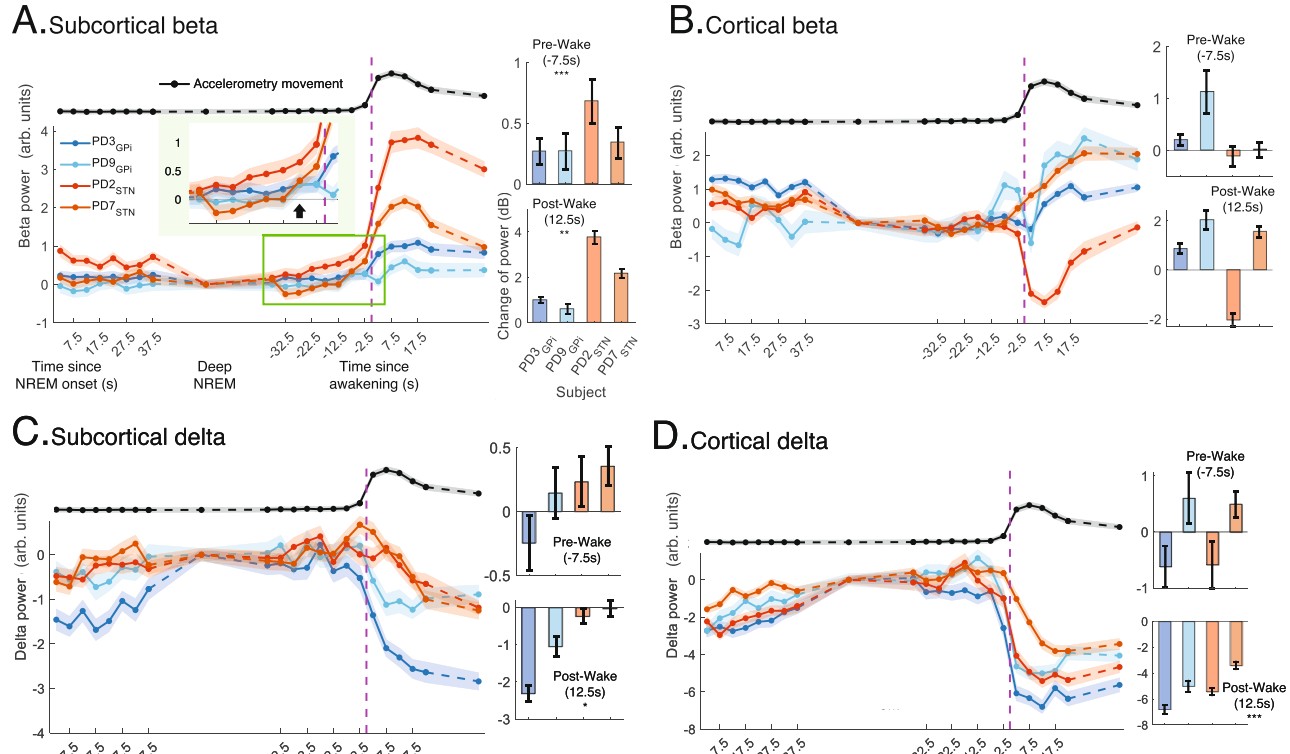

**Fig. 4 | Changes in N2/N3 NREM spectral power before spontaneous awakenings.** Subcortical beta increases before spontaneous awakening. **A** Subcortical beta power (mean ± SEM; 5 s epochs; 4 PD participants; ON stimulation; data pooled from both hemispheres) during N2/N3 NREM to wakefulness transitions (left). Vertical dashed-line (purple) shows awakening time. x-axis shows time since N2/N3 NREM sleep onset (left) and time since awakening (middle). The black line (top; Norm of RC + S accelerometry; mean ± SEM; rescaled with min-max normalization) shows across-participant movement for all N2/N3 NREM to wakefulness transitions. The bar plots (mean ± SEM) show change in subcortical beta power during immediate pre-awakening N2/N3 NREM (−7.5 s, top) and early post-awakening (+12.5 s, bottom) compared to average subcortical beta power in deep N2/N3 NREM (N2/N3 NREM data after 40 s from N2/N3 NREM onset to 40 s before awakening). Data pooled from both hemispheres. The average early post-awakening (+12.5 s; *p* value = 0.003) and immediate pre-awakening N2/N3 NREM subcortical beta

powers (−7.5 s; *p* value = 0.0003; inset zoomed plot shows the rise of beta; black arrow shows −7.5 s) are higher compared to deep N2/N3 NREM. **B** Same as **A**, for cortical beta showing no significant trend across participants for both pre and post-awakenings. **C** Same as **A**, for subcortical delta showing a significant reduction across participants for post-awakenings (+12.5 s) compared to deep N2/N3 NREM (*p* value = 0.03). **D** Same as **A**, for cortical delta power which gradually increases as sleep deepens. The average early post-awakening (+12.5 s) delta powers are lower than those during deep N2/N3 NREM (*p* value = 2.4e−20). The average early post-awakening (+12.5 s) cortical delta power is also lower than the immediate pre-awakening N2/N3 NREM delta power (−7.5 s). For barplots in all panels, LME models were constructed for bandpower with deep N2/N3 NREM vs. pre/post-awakening and disease states as fixed and participants as random effects (*n* = 1022; 5 participants) with two-sided *p* value < 0.05*, <0.01** and <0.001***. Source data are provided as a Source Data file.

accuracy; STN/GPi: 87.4 ± 3% accuracy; Fig. 5B; Table 2; Supplementary Fig. 5A and Supplementary Tables 2 and 3) for N2/N3 NREM vs. wake classification across all PD participants at the standard 30 s epoch level. We then probed N2/N3 NREM vs. wake classification at a finer temporal scale (reducing from 30 s to 5 s epoch) for rapidly dynamic awakening detection- toward sleep-specific aDBS development. Specifically, we trained participant-specific SVM models using 5 s epochs achieving high classification performances (Cortex: 90.6 ± 1.6% accuracy; STN/GPi: 80.6 ± 4.1% accuracy; Fig. 5C; Table 2; Supplementary Fig. 5B and Supplementary Tables 2 and 3) for N2/N3 NREM vs. wakefulness. This highlights the potential for rapid sleep staging using intracranial neurophysiology and faster time scales than traditional 30 s epochs. Feature ranking using mutual information revealed that the top cortical feature for N2/N3 NREM vs. wakefulness distinction was delta power while the top subcortical feature was beta power (Fig. 5D and Supplementary Fig. 5C), aligning with our earlier findings of cortical delta−subcortical beta interactions in spontaneous awakenings. Three top-ranked features for epochs near the awakening events revealed a gradual distribution of features in the manifold between N2/N3 NREM and wakefulness during the N2/N3 NREM to wake transitions (Fig. 5E and Supplementary Fig. 5D) supporting a gradual shift of the classifier output from N2/N3 NREM to awakening.

Finally, we assessed the time-resolved performance of the ML classifier models during transitions from N2/N3 NREM to wake transitions. The average wakefulness classification of the ML models during the N2/N3 NREM to wake transitions showed a gradual increase in wake detection during the awakening events (Fig. 5F and Supplementary Fig. 5F). Specifically, we found low wake detection during sustained N2/N3 NREM (sustained N2/N3 NREM: 10 ± 1.1%) and ~4x greater wake detection using cortical bandpower features preceding (*t* = −2.5 s; 37.2 ± 5.1%) the awakening events despite the low temporal resolution of the classical hypnogram (updated at a 30 s sampling rate). Classification further improved shortly after the wakening events (*t* = +12.5 s; 82.4 ± 4.7%; sustained wake: 91.56 ± 3.3%) across all PD participants. Finally, our participant-specific ML models showed consistent performance across the cross-validation schemes (2-fold vs. 5-fold; Table 2).

## Discussion

We collected multi-night intracranial cortico-basal neural recordings from five participants (four PD and one dystonia) from cortical and subcortical regions, paired with polysomnography for both DBS ON and OFF conditions, remotely in participants' own homes over 57 nights. We found increased cortico-basal slow wave and decreased beta activity as well as matching changes in cortico-basal functional

**Fig. 5 | Classification of N2/N3 NREM vs. wakefulness with cortical data.**
**A** Flowchart describing the machine learning (ML) model generation using support vector machine (SVM) and performance evaluation. **B** Performance of participant-specific ML models for N2/N3 NREM vs. wakefulness classification for all PD participants ($n = 4$) with classical 30 s epoch window in terms of confusion matrices (left) and receiver operating characteristic (ROC) performance (right). **C** Same as **B**, for 5 s epoch window. **D** Bandpower feature importance and ranking where x-axis represents 6 bandpower features and y-axis shows average mutual information between bandpower and N2/N3 NREM and wake state across all PD participants (mean ± SEM; $n = 4$; each dot is one participant). 5 s data epochs were utilized. **E** Depiction of the top three bandpower features (delta, beta and gamma) in a scatter plot for data from N2/N3 NREM to wake transitions. Data points represent 5 s epochs from a single PD participant (PD2). Color bar (left) shows the time around awakening in seconds. **F** Performance of the ML models trained on 5 s epochs shown in **C** during N2/N3 NREM to wake transitions. The x-axis represents time in seconds around awakening and y-axis is wake classification by the ML models across all transitions of the participant (mean ± SEM) with $n = 86$(PD2), $n = 104$(PD3), $n = 163$(PD7), and $n = 59$(PD9). The vertical black dashed line shows awakening time and the horizontal green dashed line represents 50% average wake detection by the models. For all panels, left and right side data were pooled. For ground truth of 5 s epochs, actual awakening events within the classical 30 s sleep epochs were determined with EEG and accelerometry data by a board certified sleep physician and then segmented into N2/N3 NREM and Wake 5 s segments (see "Methods"). Source data are provided as a Source Data file.

**Table 2 | N2/N3 NREM vs. wakefulness classification in PD participants**

| | Cortex | | | | Subcortex (STN/GPi) | | | |
|---|---|---|---|---|---|---|---|---|
| | 30 s epoch | | 5 s epoch | | 30 s epoch | | 5 s epoch | |
| | 5-fold CV | 2-fold CV | 5-fold CV | 2-fold CV | 5-fold CV | 2-fold CV | 5-fold CV | 2-fold CV |
| Accuracy | 93.60 ± 1.40 | 93.65 ± 1.37 | 90.60 ± 1.63 | 90.93 ± 1.61 | 87.43 ± 3.04 | 87.15 ± 3.09 | 80.62 ± 4.08 | 81.80 ± 3.62 |
| AUC | 97.50 ± 0.67 | 97.53 ± 0.69 | 95.42 ± 1.10 | 95.70 ± 1.08 | 94.02 ± 1.83 | 93.93 ± 1.85 | 88.30 ± 3.37 | 89.23 ± 2.97 |
| Sensitivity | 93.88 ± 1.67 | 93.50 ± 1.14 | 93.62 ± 1.23 | 90.70 ± 1.47 | 86.50 ± 3.26 | 88.00 ± 2.93 | 88.72 ± 1.36 | 82.48 ± 4.66 |
| Specificity | 93.30 ± 1.15 | 93.88 ± 1.60 | 87.57 ± 2.06 | 91.13 ± 1.79 | 88.35 ± 2.85 | 86.33 ± 3.25 | 72.53 ± 8.14 | 81.08 ± 2.86 |
| PPV | 93.35 ± 1.19 | 93.85 ± 1.58 | 88.32 ± 1.87 | 91.13 ± 1.74 | 88.07 ± 2.93 | 86.55 ± 3.17 | 77.38 ± 5.13 | 81.18 ± 3.11 |
| NPV | 93.88 ± 1.63 | 93.48 ± 1.17 | 93.20 ± 1.38 | 90.73 ± 1.48 | 86.80 ± 3.17 | 87.75 ± 3.02 | 86.22 ± 1.70 | 82.50 ± 4.16 |

Individual machine-learning model performance (mean ± SEM) for N2/N3 NREM vs. wakefulness binary classification using cortical data.
*PPV* positive predictive value, *NPV* negative predictive value, *AUC* area under the receiver operating characteristic curve, *CV* cross-validation.

connectivity during N2/N3 NREM, an effect that was enhanced by DBS. Within N2/N3 NREM, there was a direct inverse relationship between subcortical beta and cortical delta activity and further, we found that subcortical beta power rose prior to spontaneous awakenings at high temporal resolution (5 s). These data support the hypothesis that subcortical beta is related to overnight sleep disruptions and spontaneous awakenings in PD. Finally, we utilized ML models on cortico-basal intracranial data and achieved high performance in classifying N2/N3 NREM and wakefulness both in the classical (30 s) and in subclassical rapid (5 s) time windows, providing a foundation for future personalized sleep adaptive DBS.

It is established that during the daytime, subcortical beta oscillations are excessive in PD and potentially contribute to circuit disruption and motor symptoms[31,32]. Here, we show that subcortical beta oscillations also disrupt cortical slow oscillations during N2/N3 NREM sleep in humans with PD and are partially responsible for awakenings during the night, validating findings from PD models in primates[17]. The rise of subcortical beta at least 7.5 s before awakenings, at which time delta didn't yet show a statistically significant change, plus the leading of subcortical beta (9 s–4 s; 3 out of 4 PD participants; Fig. 3D) compared to cortical delta within N2/N3 NREM, support a potential causal relationship between awakenings and subcortical beta in PD. Sleep-related cortico-basal network beta fluctuations also have translational implications for emerging adaptive DBS therapies that often use beta as a control signal input. The overall reduction in beta as shown in our participants would result in a down titration of stimulation in beta triggered aDBS algorithms which could be problematic if higher stimulation amplitudes are beneficial for sleep in a given patient. Conversely, the rapid rise of the subcortical beta prior to awakenings should, in theory, trigger an increase in stimulation amplitude that could be beneficial. Although, the response to beta amplitude would be dependent on the parameterization of the control algorithm and the rapidity of stimulation amplitude changes, with opportunity for personalization. Also, the subcortical beta signal, although captured by the statistical (LME) models, was also partially obscured by a lower SNR in some patients, which potentially poses a challenge for ML algorithms to track and utilize this biomarker for online pre-wake prediction. Further, we show that DBS stimulation, while known to reduce subcortical beta oscillations during wakefulness[33], also significantly impacts cortical delta and low-beta power during N2/N3 NREM sleep. This finding aligns with previous studies where an increased accumulation of EEG delta power during NREM sleep was found as a result of subthalamic DBS in PD[34], but here provides a candidate causal mechanism. Data in our study indicates that DBS therapy appears to improve sleep in PD, at least in part, through direct modulation of beta and delta oscillations.

The link between sleep dysfunction and daytime motor, mood and cognitive symptoms makes sleep an enticing potential target for further investigation[5–7]. Moreover, sleep disturbances, and particularly reductions in cortical slow wave activity during NREM have been linked to faster disease progression[11,14]. Therefore, improving NREM sleep architecture with adaptive DBS has the potential to reduce overnight insomnia, improve waking motor and non-motor symptoms and increase cortical slow waves that could impact disease progression. This supports the proposal that daytime neural activities and overnight sleep physiology are notably dissociable and require different strategies for aDBS to optimize rhythms during these two distinct phases. Implementing different aDBS algorithms around the circadian cycle could be achieved by the introduction of daytime (versus sleep) neural classifiers, circadian (clock) based algorithms and combined feedforward and feedback controllers that optimize both daytime and nighttime neurophysiology[35,36].

Our ML analyses demonstrate N2/N3 NREM vs. wakefulness classification not only in classical 30 s sleep epochs but also at rapid time windows (5 s) with high accuracy. The ML models showed increasing wake detection around the actual awakening events using intracranial brain recordings (Fig. 5F). The performance of these ML models, despite being constrained to simpler ML algorithms (with potential for embedding on emerging DBS devices) and having only limited power band feature inputs, suggests the viability and potential applications of machine-learning algorithms for identifying micro-stages of sleep and designing adaptive DBS therapies that can modulate stimulation to manage or prevent awakenings.

Limitations include the fact that our ground-truth sleep stage labelings were obtained through a portable polysomnogram and automated sleep-scoring algorithm, validated on healthy controls[26], instead of a conventional laboratory-based PSG. However, we note that our intracranial recordings, grouped according to sleep stages defined from our portable PSG, revealed anticipated and classical changes in cortical (ECoG) activities across various stages (Fig. 1F, G and Supplementary Fig. 4). In particular, the observed elevation in cortical delta power during N3 sleep and reductions in beta power provide evidence of the differentiation of underlying sleep stages within our group of participants using this pipeline (Fig. 1F, G and Supplementary Fig. 4). To mitigate any limitations of the automated sleep-scoring algorithm, we manually re-scored all awakenings by a board certified sleep physician. Furthermore, our portable remote setup enabled us to collect multi-night recordings in a natural setting which compares favorably to single-night PSG recordings (from a sleep laboratory) that can be subject to first night acclimatization and sleep disruption effects. We also report many nights of recordings per participant (n = 57 total), but from a relatively small number of participants, which supported highly statistically powered LME analyses that modeled within, as well as across, participant effects—similar to the strengths of primate research. This approach was found to be well suited to looking for the within-participant cortico-subcortical interactions which were the primary focus of this study. However, evaluation of across participant factors, including analysis of how beta-delta interactions predict

clinical outcomes at scale would require a different study design with large numbers of participants (but would also require much less within-participant data). Finally, our comparison participant was a single cervical dystonia patient (rather than a formal control group) reflecting the uniqueness of this participant cohort, with high-resolution sensing-enabled pulse generators and chronically implanted ECoG electrodes. However, despite this, and in view of the large within-participant dataset size and LME modeling, we were able to find exploratory evidence in support of a difference between the dystonia participant and the PD group, which motivates future larger studies with formal comparison at the group level. In this study, we restrict our analysis to NREM and canonical power bands with a focus on beta and delta[30]. We did not examine changes in other sleep stages or specifically analyze sleep spindles (which overlap in frequency with low beta) or other frequency bands. In particular, PD is associated with REM sleep behavior disorder and a detailed analysis of the changes in neurophysiology during REM in PD might also reveal PD-specific cortico-basal neurophysiology which will be addressed in future analyses. The setup of this study, albeit naturalistic (at-home multi-night recordings), did employ investigational DBS hardware including the neurostimulator (Summit RC + S) and chronic cortical electrocorticography that are not widely available. This was necessary to elucidate the network-level mechanisms of sleep disruption in PD. However, the identified biomarker, subcortical beta, could be targeted using currently available devices and electrodes. Finally, our ML models in this study were limited to straightforward binary classification (N2/N3 NREM vs. wakefulness) and were region-specific (trained separately for cortical and subcortical data) with a view toward the constraints of current and emerging sensing-enabled DBS devices. While we could not conduct out-of-sample tests due to the limited availability of data from similar setups, we utilized multiple cross-validation schemes for validating the performance of the ML models.

One of the major technical aspects of our study was recording full-spectrum, time-domain, intracranial cortical and subcortical neural activity during sleep, over multiple nights ($n = 57$). This research leveraged the availability of a critical investigational DBS device (Summit RC + S), which is no longer available. This study data and that by other teams in our research community supports the need for such ongoing DBS research tools to be made available through collaboration between academics, regulators and industry[37].

In this study, we recorded and analyzed intracranial oscillatory neural activity with extracranial polysomnography at-home over multiple nights in PD participants, in the presence and absence of DBS. Our data revealed that cortico-basal network spectral power and connectivity in the delta and beta bands are increased and decreased in N2/N3 NREM versus wakefulness respectively, an effect that was enhanced by DBS. Further, within N2/N3 NREM, cortical delta band slow wave activity was inversely related to subcortical beta, which also rises prior to spontaneous awakenings. Finally, our machine-learning models achieved high accuracy in distinguishing between N2/N3 NREM and wakefulness, both in classical and a faster time scale. These findings uncover a role of subcortical beta in sleep dysfunction in PD and provide targets as well as machine-learning models for future personalized sleep-specific adaptive DBS.

## Methods
### Participants, demography, and ethics
This study was reviewed by our Institutional Review Board (University of California San Francisco Institutional Review Board) and registered on clinicaltrials.gov (NCT03582891; IDE G180097). The study was also reviewed by the Human Resources Protection Office (HRPO) at Defense Advanced Research Projects Agency (DARPA). This study was conducted in accordance with the Declaration of Helsinki. All participants provided informed written consent for participation in the study and publishing of their de-identified data. No direct compensation was

provided to the participants for participating in the study. However, participants were reimbursed for mileage driven from home and hotel fees for study visits. We recruited 4 participants with idiopathic PD for this study (Table 1). A movement disorders physician diagnosed each individual with PD according to the Movement Disorder Society PD diagnostic criteria[38]. The motor component of the United Parkinson's Disease Rating Scale (UPDRS) scores were administered by trained raters. We also recruited one participant with cervical dystonia as a comparison participant. Participants were recruited from a parent study focused on investigating closed-loop DBS for daytime motor symptoms. Details of the study protocol are included in the Supplementary Materials under "Study protocol documentation" section. Implanted electrodes were connected to an investigational sensing-enabled Summit RC + S DBS implantable pulse generator provided by Medtronic (Fig. 1A)[24]. The First participant enrolment for this study was in February 2021 and the last enrolment was in June 2021. All participants had chronic bilateral cortical ECoG electrodes and two PD participants were implanted with bilateral electrodes in the Subthalamic Nucleus (STN; PD2 and PD7) and two PD participants along with one dystonia participant were implanted with bilateral electrodes in the Globus Pallidus (GPi; PD3, PD9 and dystonia participant) nuclei (Fig. 1B). DBS electrode implantation targets were determined by the clinical team. A movement disorder specialist programmed the participants with conventional DBS settings, optimizing stimulation to address daytime motor symptoms. The cohort investigated here included by definition patients undergoing DBS implantation. This introduces a potential bias regarding the severity of the disease (PD/cervical dystonia) studied by including only those with relatively more advanced disease conditions.

### Experimental design and protocols
We collected data using two protocols: long-term multi-night data collection ON stimulation plus separate two night comparison recordings, one night ON DBS and one night OFF DBS. The primary outcome of the study was the interaction between subcortical beta and cortical delta. The secondary outcome of the study was the elevated amplitude of beta prior to spontaneous awakenings. During the long-term overnight data collection, each participant ($n = 5$) was equipped with a portable PSG (Dreem2) headset and overnight intracranial data as well as polysomnography data were recorded for ~10 nights (Supplementary Table 1) that were predominantly consecutive. We initially collected a total of 58 nights of data. Upon manual inspection, we rejected a recording from one night which had 95% missing intracranial data. In the ON/OFF protocol, overnight data from the PD participants ($n = 4$) were collected for two consecutive days. On the first day DBS was ON ($3.075 \pm 0.65$ mA) and the next day DBS was OFF. During both data collection protocols, the PD participants were on their regular clinical dopaminergic replacement medications. ON/OFF recordings were not completed in the cervical dystonia participant at the participant's request. All data recordings were performed remotely in participants' homes.

### Polysomnography acquisition
Extracranial polysomnography (PSG) was recorded through the Dreem2 headband which includes an automated sleep staging algorithm with extracranial electroencephalography (EEG) data (Dreem2 headband, Dreem Co., Paris, France)[26,27]. The Dreem2 headband provided sleep stage classification hypnograms according to AASM scoring methods[26,27]. The sleep staging was performed using EEG data at every 30 s epoch. Sleep onset was defined as the start of the NREM sleep (3 consecutive epochs were required to classify N1). Wakefulness after sleep onset (WASO) was calculated as the total waking time after sleep onset and before the last epoch of sleep. Finally, to overcome any loss of accuracy incurred through automated sleep staging and to obtain a more precise timing estimate of awakenings (automated sleep

staging works on a 30 s epoch window), we manually re-scored the N2/N3 NREM to wakefulness transitions through evaluation of the portable PSG recordings (EEG and accelerometry data), according to American Academy of Sleep PSG staging rules. Manual scoring was taken as primary where there was disagreement between manual and automated scores.

## Intracranial data collection

For each participant, the Summit RC + S device was implanted bilaterally and connected to bilateral sensing and stimulation-capable quadripolar leads in the basal ganglia targets (STN in 2 PD participants or GPi in 2 PD participants and 1 cervical dystonia participant) plus quadripolar sensorimotor chronic electrocorticography (ECoG), sensing only strips, with 4 electrode contacts spanning the central gyrus (Fig. 1B). Overnight intracranial data were collected from cortical and subcortical structures in both left and right hemispheres (Fig. 1D) in addition to data from bilateral accelerometers embedded within the chest-mounted pulse generator devices. The time series FP data were recorded at either a 250 Hz or 500 Hz sampling rate.

## Data preprocessing

The intracranial recordings were validated and synchronized to the PSG recordings using accelerometry data. Cross-correlation was applied to accelerometry data from both the Dreem2 band and the RC + S neurostimulator in order to ascertain the delay between PSG and RC + S time series (Supplementary Fig. 3). All intracranial data were downsampled to 250 Hz and filtered through a 0.8–100 Hz zero-phase IIR elliptic bandpass filter with 1 dB passband ripple and 100 dB attenuation ("filtfilt" and "designfilt" function in Matlab; Fig. 1E). Large artifactual spikes in the subcortical intracranial data were removed along with the corresponding cortical data (Supplementary Fig. 3). To identify artifacts, absolute squared subcortical data were first smoothed with a Gaussian kernel with 1 s window then any period larger than 5 times the median over the whole night was considered artifactual spikes. The subcortical data underwent artifact removal for ECG interference through the application of an optimized combination of two complementary ECG data removal algorithms ("PerceptHammer" and "Perceive" library; Matlab; Supplementary Fig. 6)[28,29]. For each 10 min non-overlapping window of subcortical FP data, the "Perceive" library[28] was first utilized to generate an initial starting seed of ECG artifact template based on the presence of characteristic sharp QRS-like signal deflections. This was done to personalize the initial seed template and account for the variations of ECG artifacts that occur across hemispheres (left/right side), participants and even duration of the night. For each night, all ECG artifact templates for 10 min windows found by the Perceive method were averaged which was then fed to the Template subtraction pipeline (PerceptHammer[29]) as an *initial* template seed for the actual ECG artifact detection and removal operation. Notably this second algorithm uses Woody's adaptive filter to identify locations of the artifact and then update the template recursively to improve it further. The template subtraction pipeline was applied separately for each 10 min non-overlapping window with the same initial template seed. This window-wise ECG removal was implemented to account for any changes of ECG artifact throughout the night. Finally, forced searches were conducted by the PerceptHammer pipeline for artifacts missed by the adaptive filter. In order to avoid the template locking into a low-frequency rhythmic neural activity during the recursive update which results in the removal of low-frequency contents of neural activities, we compare the final template formed by Woody's adaptive filter with the initial template seed generated by the Perceive method via normalized cross-correlation. If the maximum cross-correlation is less than a predetermined threshold of 0.9, the results were rejected and the PerceptHammer pipeline was re-applied via forced searches without any recursive update of the initial template.

## Power spectrum analysis

To calculate the power spectra, the intracranial data from each night were z-scored for each location. Then, the N2/N3 NREM data segments were collected together according to the PSG hypnogram labels. The selected data were segmented into 5 s epochs and power spectra were calculated for each epoch using a Hamming window of 1 s, 512-point FFT with 50% overlap by Welch's method ("pwelch" in Matlab) which was normalized by the total power in 0−50 Hz. The calculated power spectrums for each epoch were then pooled over both hemispheres within participants. For calculating the change in power spectrum in N2/N3 NREM with wakefulness as the baseline, the power spectrums for wake epochs were calculated in a similar manner as during N2/N3 NREM and the difference between the average wake power spectrum and N2/N3 NREM power spectrum for each night was calculated. For calculating the ON vs. OFF power spectrum, average power spectra were calculated for ON and OFF nights and their difference was taken. The averages were calculated on log-transformed power spectra.

## Spectral coherence analysis

To compute the spectral coherence, the intracranial data obtained from each night were normalized using z-scoring for each location. Subsequently, N2/N3 NREM data segments were extracted and then divided into 5 s epochs. For each epoch, 5 s of cortical and subcortical data were utilized for estimating the one-sided magnitude squared coherence using the multitaper method ("mscohere"; Matlab) with Hamming window of 1 s and 512-point FFT. The epoch-wise spectral coherences were then pooled over both hemispheres. Similarly to the power spectral analysis, spectral coherences for wake epochs were calculated and the difference between the average of wake and N2/N3 NREM spectral coherence for each night was calculated in order to obtain the change in spectral coherence in N2/N3 NREM with wake as the baseline.

## Beta-delta correlation analysis

To analyze the interaction between subcortical beta and cortical delta activity during N2/N3 NREM sleep in intracranial signals, we applied z-scoring, power spectrum calculation and normalization techniques as previously described. However, there was one exception regarding the normalization of the cortical power spectrum where instead of normalizing it by dividing the total power (0–50 Hz), we divided it by the total power excluding the beta range (0–13 Hz and 31–50 Hz). This adjustment was necessary to avoid detecting spurious negative correlations that could be caused by the normalization procedure itself. However, we note that both normalization methods (with and without excluding beta range during normalization) provided highly similar statistical outcomes and conclusions. Both subcortical beta and cortical delta were calculated for 5 s epochs which were log-transformed for each night and each hemisphere. The band powers were then pooled over both hemispheres. Subsequently, for each participant, we calculated the Spearman's rho correlation coefficient between subcortical beta and cortical delta power across all 5 s epochs for each night. For calculating the delay between subcortical beta and cortical delta power, normalized cross-correlation ("xcorr" function in Matlab) was calculated between these band powers of the 5 s epochs from above for each night. Lag was calculated by finding the minimum (trough, reflecting a negative relationship) normalized cross-correlation between the two band powers. Epoch band powers for each night were smoothed using a 20-point Gaussian kernel. Data for each night were mean-subtracted and pooled from both hemispheres. To investigate interactions between delta and beta powers from cortex, we applied the same power spectrum calculation techniques on 5 s epochs as previously described in beta-delta correlation analyses. The only exception was the normalization step of the power spectrum which was not applied to avoid detecting artificial negative correlations that could be imposed by the normalization of the power spectrum. In order to avoid any risk of introduction of a potential spurious negative interaction in our single

(cortical) site analysis we removed the normalization step for investigating cortical-cortical delta-beta interactions. Spearman's rho rank correlation coefficient was calculated between the cortical delta and beta power in all 5 s epochs throughout each night for all participants.

## NREM to wake after sleep transition analysis

To investigate the changes in spectral power during N2/N3 NREM to wake transition events, all intracranial data were bandpassed using the zero-phase IIR elliptic bandpass filters. Next, the data were z-scored for each night for each hemisphere at all locations. Hilbert transform was applied to the z-scored data and the absolute squares of the results were converted into a decibel scale for band-specific power. After detecting all N2/N3 NREM to wake transitions, events with N2/N3 NREM sleep less than 85 s and wake periods of less than 25 s were ignored. Band-power for each 5 s epoch was averaged. The time label for each 5 s epoch was found by taking the average of the start time and stop time of the epoch (e.g., epoch from 0 s to 5 s was labeled as 2.5 s). All epochs in N2/N3 NREM data after 40 s from N2/N3 NREM onset and 40 s before awakening were averaged to calculate deep N2/N3 NREM (slow-wave sleep; SWS) power. All epochs in awake stage data after 25 s from wake onset were averaged to calculate awake stage power. All data were analyzed from the ON stimulation multi-night dataset.

## Machine-learning models

For both 30 s and 5 s epoch windows, individual SVM models ("fitcsvm" in Matlab) with Gaussian kernel were trained for each PD participant. For class balance, we randomly reduced the number of N2/N3 NREM epochs (using "datasample" in Matlab) to match the total number of wake epochs. To calculate the bandpower features, we first standardized the intracranial data for each location across each night using z-score. Then, the N2/N3 NREM and wake stage data were collected based on the PSG hypnogram labels. These data were segmented into 30 s and 5 s epochs and power spectra were calculated for each epoch using Welch's method with a 1 s Hamming window and a 512-point FFT, along with a 50% overlap ("pwelch" in Matlab). Uniform prior distribution was assumed during training and no score transform was applied. Kernel width was 2.6 and the standardization of the features were enabled. 5-fold cross-validations were performed for observing the N2/N3 NREM vs. wake classification performances. For the models using 5 s epochs, customized cost function was applied during the training to make the models more sensitive to wake events. The threshold for binary classification was 0.5 and no further threshold optimization was conducted. For feature importance and ranking, the mutual information between bandpower features of 5 s epochs and N2/N3 NREM vs. wake class labels were calculated[39]. Similar feature ranking results were obtained using 30 s epochs. Data from left and right hemispheres were pooled for all analyses. In addition to 5-fold cross-validation, we also implemented a 2-fold (50% of the data were used for training and the remaining data were tested) cross-validation scheme to investigate the robustness of our performance.

## Statistical methods

A significance threshold of 0.05 was employed to determine statistical significance. Linear mixed effect models (LME) were utilized ("fitlme" in Matlab) for investigating the spectral power and coherence differences, the interactions between cortical and subcortical beta with cortical delta powers. For LME models, full covariance matrix was applied using the Cholesky parameterization which estimated all elements of the covariance matrix. We accounted for the correlation between 5 s epochs within the same night by including "night #" as a random effects term in our LME models. Theoretical likelihood ratio test ("compare" in Matlab) was used for comparing LME models. Wilcoxon rank sum test ("ranksum" in Matlab) was utilized for measuring group-level differences in wake predictions. All analyses were performed using Matlab 2021b (Mathworks).

## Reporting summary

Further information on research design is available in the Nature Portfolio Reporting Summary linked to this article.

## Data availability

All data supporting the findings of this study are available within the article and its Supplementary files. Any additional requests for information can be directed to, and will be fulfilled by, the corresponding authors. The raw (identifiable) data from participants are privacy-protected. The processed de-identified data can only be shared on request as these datasets are part of an ongoing medical research study with active data collection and will subsequently be shared according to the National Institute of Health (NIH) data sharing requirements. Source data are provided with this paper.

## Code availability

Codes generated and/or analyzed in this study will be shared upon reasonable request for reviewing purposes only. These codebases are part of the analyses of an ongoing medical research study. Publicly accessible code will be available upon the completion of the ongoing study at: https://github.com/MDFahimAnjum.

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

## Acknowledgements

This work was partly funded by the Defense Advanced Research Projects Agency (DARPA) under grant HR0011-20-2-0028 awarded to T.D. and S.L. and subsequently by the National Institute of Neurological Disorders and Stroke (NINDS) under grant 1R01NS131405-01 awarded to S.L. Google Research provided unrestricted laboratory gift awarded to S.L. to support for our laboratory. We express our gratitude to the participants for their valuable contributions to this study. DARPA, the NINDS and Google research played no role in the study design, data collection, analysis, and interpretation of data, or the writing of this manuscript. We thank and acknowledge Shravanan Ravi for his help supporting data collection and Tom Wozny for his contributions in visualizing ECoG and DBS lead locations in Fig. 1B. We also thank and acknowledge Kenneth X. Probst for his contributions to Fig. 1A, 1C. Thanks to all the participants who joined the study and to the staff in the Department of Neurology of the University of California San Francisco for their help and support.

## Author contributions

S.L. and M.F.A. designed the study and outlined the data analysis and statistical models; C.S. and M.F.A. collected and processed data; M.F.A. and C.S. designed and developed software; P.S. and S.L. managed patients and surgeries; R.Z. manually rescored the polysomnography data; M.F.A. analyzed the data, conducted statistical tests, prepared models, and drafted the manuscript; S.L. and T.D. acquired funding; S.L. supervised the research; D.J.D and T.D. provided input on data interpretation; all authors reviewed, edited, and approved the manuscript.

## Competing interests

S.L. has received honoraria from Medtronic and is a paid consultant for Iota Biosciences. T.D. is founder-chairman of MINT neurotechnology, founder/CSO of Amber Therapeutics (bioelectronic medicines), and a paid advisor for Cortec Neuro. T.D. has research collaborations with Magstim Ltd, Medtronic, and Bioinduction Ltd. S.L., C.S., P.S., M.F.A, and T.D. are involved in a pending patent application. Patent applicant: University of California San Francisco (UCSF); Inventors: S.L., C.S., P.S., M.F.A., and T.D.; Application number: 63/522,284; Status: Provisional; Specific aspect: Awakening detection and intracranial machine learning models for detection of awakenings.
