## [Peer Review File · Nature Communications]

REVIEWER COMMENTS

Reviewer #1 (Remarks to the Author):

Thank you for inviting me to review this interesting study. In the present article, Anjum et al. studied sleep related changes and deep brain stimulation related (DBS) modulation of cortico-basal ganglia oscillations in four patients with Parkinson's disease and one subject with dystonia. The results demonstrate that beta and delta rhythms in invasive brain activity relate to sleep disturbances, a significant burden to patients. Moreover, this is the first study to highlight the modulatory effect of DBS on these oscillatory changes, opening new horizons to sleep stage specific neurotherapies in the treatment of Parkinson's disease. The key limitation of the study is the low patient count, with only five subjects reported. However, this is compensated by the level of detail and amount of data that the authors have acquired through their unique neurotechnological approach. In this approach also lies the key novelty which allowed to characterize sleep data in at-home settings in ~500 hours of invasive recordings from both cortex and subcortex. The study has broad implications for fundamental sleep physiology, pathophysiology of PD and the development of neuromodulation therapies. Beyond this, it highlights the importance of brain rhythms and their interareal synchronization for fundamental insights into human brain function and the development of clinical brain computer interfaces.

The following comments aim to help to further improve the manuscript:

1. It is unclear, why the others chose to leave out REM sleep stages in the detailed analysis, especially as PD is associated with REM sleep behavior disorder. This should be added. If not possible, e.g. because REM is less precisely identified by the Dreem hardware, this should be highlighted and the wake-up analysis should be discussed potentially more cautiously as these sections could be REM.
2. The authors note that cortical beta power was higher in PD than in the dystonia patients. This is interesting, because the question whether cortical beta is exaggerated in PD remains a topic of debate. I would like to invite the authors to discuss this in a bit more detail, comparing wake, REM and NREM in this cohort.
3. Given the importance of it, I would suggest to add brief mention how extracranial and intracranial signals were synced and how artifacts were removed in the first paragraph of the results text, as this is critical to judge the validity of the subsequent sections for the reader.
4. Similarly, describe that sleep stages were defined based on Dreem before the first mention of the results with respect to sleep stages. Consider a half sentence on performance / validation there.

5. Given the fact that subcortical beta activity is a major hallmark of PD and that non-human primate work that in part inspired this study have mostly focused on beta activity, it was sometimes confusing that the authors chose to highlight changes in delta first, especially in the early results section. I would suggest to consider reporting subcortical and cortical beta first as the hypotheses and expected results can be more easily followed by the reader and therefore may improve readability and match expectations better.

6. From my perspective the discussion can be improved. I personally believe that hinting to limitations of previous studies as done in paragraphs 2-4 of discussion is better shortened and shifted to introduction to be rephrased as unmet needs. I got bored reading what others haven't done instead of learning how the authors conceptualize their findings. I guess this is rather an artifact of editorial strategies, as journals increasingly prioritize reviewers comments or ratings with respect to novelty, but often fail to find reviewers who can actually judge the novelty sufficiently. Because of this increasing trend it is understandable that authors devote a large section of text to highlighting the novelty. Here is an expert reviewer who can and does identify this study as clearly novel, so maybe this paragraph just isn't for me. In case a revision is invited, perhaps this section can be replaced by a thorough discussion of the impact of the present study on the current ongoing development of closed-loop neurostimulation therapies for PD and other brain disorders. Specifically, it would be important to discuss the implications for the ongoing ADAPT-PD trial on beta based adaptive DBS. Furthermore, it should be discussed to what extent cortical signals will be a key for successful closed-loop therapies in PD. I also found the finding that subcortical beta precedes cortical delta very interesting and deserving of more detailed discussion.

7. The authors state that the challenge to record invasive neurophysiology is "solved" by sensing enabled devices. I understand that this is strictly speaking true, but it comes with a more general notion that it is no longer difficult to obtain naturalistic data, which is absolutely not true. Therefore, I feel that this sentiment is a massive overstatement that is potentially harmful to future studies that overcome the difficulties of gaining access, overcoming regulatory burdens, developing computational tools and identifying patients who are willing to participate in research with such devices. Recording invasive neurophysiology in naturalistic settings will be a significant challenge for at least another five years. In fact, the authors will very likely not be able to perform this kind of research in five years, because the device and the ECoG strip electrode is no longer supported by manufacturers with currently no real viable alternative in sight.

8. For future submissions, please do the reviewers the favor of adding figures directly in text, even if the journals don't want that. It makes reviewing so much more joyful.

9. The "FP" abbreviation makes it unnecessarily difficult to read, I would suggest to get rid of it. Use activity, oscillations or any other generic term that is readable.

10. I would suggest to add a table for the machine learning performances in main text, as the results get to be a bit convoluted.

11. Fig. 2 D subcortex one signal looks (PD3 GPi) odd, did the authors look at the change in wakefulness to rule out stimulation induced artifacts?

12. In Figure 3 B OFF Figure 4 box plots individual data points are missing, please add or change plot type.

13. Add spearman's rho values and p-values to scatter plots in Figure 3 as the massive point clouds are hard to interpret.

14. Add description/legend to black line to Figure 4 inside panels A-D

Reviewer #2 (Remarks to the Author):

The paper of Anjum et al. explores intriguing aspects of subcortical and cortical oscillations during sleep, shedding light on major scientific findings and potential clinical applications. There significant discovery is the observed correlation between delta and beta connectivity during sleep, where delta increased and beta decreased, showcasing enhanced coherence under deep brain stimulation.

The clinical translation of this finding is noteworthy, particularly the ability to predict NREM N2/3 sleep versus wakefulness within a short 5-second window. While the paper holds promise for the broad readership of the journal, several critical points require attention, including the need for direct testing of translational capabilities and addressing methodological concerns.

Major points of concern:

i. The impact of medication withdrawal ("Med off") on delta-beta coherence raises questions about the differential effects of dopamine and DBS. A detailed exploration of how DBS acts similarly or differently compared to dopaminergic interventions is crucial for a comprehensive understanding.

ii. The limited sample size of only four subjects is a notable concern. A power analysis is essential to evaluate the stability of findings across subjects. Additionally, distinctions between the effects of DBS on two different target areas (2 GPi and 2 STN) should be explored, considering potential variations in response.

iii. Relying on a single dystonia patient as a control may not provide sufficient grounds for drawing meaningful conclusions. A more diverse control group is needed to strengthen the validity of the findings and draw accurate comparisons. As an alternative to dystonia patients, a dataset from patients with epilepsy undergoing stereo-electroencephalography (SEEG) could be considered.

iv. The predictive performance of the N2/3 NREM sleep prediction (5-second recording) holds translational value for adaptive applications. However, to establish robust predictive capabilities, it is imperative to demonstrate performance across an out-of-sample cohort comprising a minimum of 15-20 subjects. This would enhance the generalizability and reliability of the proposed prediction model.

v. The 'perceive' library's ECG removal technique using QRS interpolation, contrary to the method described in the Hammer et al. 2022 paper, fails to effectively eliminate T waves in the ECG signal, leaving behind residual low-frequency artifacts. Even employing alternate methods such as SVD or template subtraction leads to the inadvertent removal of low-frequency power, resulting in a reduction of spectral content. Given the relevance of delta bands in this study, the approach to mitigate this discrepancy remains unclear. To gain a comprehensive understanding, it would be beneficial to visually compare the power spectrum pre- and post-ECG removal and should be presented.

vi. Under the 'Power Spectrum Analysis' section, the normalization of power spectra using the total power within the 0-50Hz range is specified. However, in the subsequent 'Beta-Delta Correlation Analysis,' the exclusion of beta power (14-30 Hz) in subcortical beta and cortical delta is puzzling. This omission could potentially skew the distribution and bias the correlation towards the beta band, impacting the accuracy of the results. The rationale behind this exclusion lacks sufficient explanation and may introduce distortions in the findings.

Additionally, the use of a log transform in the analysis complicates the interpretation of the resulting spectrum by compressing higher frequency bands more than lower frequency ones, introducing complexity into the interpretation process.

Moreover, during the correlation assessment between cortical beta and delta, the absence of power spectrum normalization adds to the inconsistency observed in the methodology. This non-uniformity might significantly influence the obtained results, warranting a clear justification for its presence in the analysis.

Reviewer Responses - Multi-night cortico-basal recordings reveal mechanisms of NREM slow wave suppression and spontaneous awakenings at high-temporal resolution in Parkinson's disease

We are grateful to the reviewers for their insightful and helpful comments that we have used to improve the manuscript substantially. We provide detailed responses to each suggestion below as well as key changes to the text.

- Reviewer comments in plain text
- *Author responses in italic*
- **Changes in the manuscript in bold**

Reviewer #1 (Remarks to the Author):

Thank you for inviting me to review this interesting study. In the present article, Anjum et al. studied sleep related changes and deep brain stimulation related (DBS) modulation of cortico-basal ganglia oscillations in four patients with Parkinson's disease and one subject with dystonia. The results demonstrate that beta and delta rhythms in invasive brain activity relate to sleep disturbances, a significant burden to patients. Moreover, this is the first study to highlight the modulatory effect of DBS on these oscillatory changes, opening new horizons to sleep stage specific neurotherapies in the treatment of Parkinson's disease. The key limitation of the study is the low patient count, with only five subjects reported. However, this is compensated by the level of detail and amount of data that the authors have acquired through their unique neurotechnological approach. In this approach also lies the key novelty which allowed to characterize sleep data in at-home settings in ~500 hours of invasive recordings from both cortex and subcortex. The study has broad implications for fundamental sleep physiology, pathophysiology of PD and the development of neuromodulation therapies. Beyond this, it highlights the importance of brain rhythms and their interareal synchronization for fundamental insights into human brain function and the development of clinical brain computer interfaces.

We thank the reviewer for this summary and for highlighting the very large within-subject dataset that was leveraged for this analysis.

The following comments aim to help to further improve the manuscript:

1. It is unclear, why the others chose to leave out REM sleep stages in the detailed analysis, especially as PD is associated with REM sleep behavior disorder. This should be added. If not possible, e.g. because REM is less precisely identified by the Dreem hardware, this should be highlighted and the wake-up analysis should be discussed potentially more cautiously as these sections could be REM.

This is a very insightful comment. Indeed, PD is associated with RBD plus interesting neurophysiological changes during REM. We agree that the changes in neurophysiology during

REM warrant a detailed analysis (which is in progress), however, like the NREM analysis covered here, the REM analysis is very extensive and requires its own detailed and dedicated manuscript, which we will present next year. We now have discussed these points in the Discussion and added the lack of REM analysis to the limitations of our study:

We did not examine changes in other sleep stages or specifically analyze sleep spindles (which overlap in frequency with low beta) or other frequency bands. In particular, PD is associated with REM sleep behavior disorder and a detailed analysis of the changes in neurophysiology during REM in PD might also reveal PD specific cortico-basal neurophysiology which will be addressed in future analyses.

Additionally, we agree with the reviewer about the possible limitation of Dreem hardware for detecting REM which can potentially affect the analyses of awakenings. However, we note that in the validation of the Dreem polysomnography using healthy controls, the difference in accuracy of the automated hypnogram and manual PSG scoring consensus was small (84.5% accuracy of Dreem vs 87.8% accuracy of manual scoring in REM detection¹). Nonetheless, in light of these comments, we now have manually re-scored all of the PSG awakenings with the input of a board certified sleep expert (Dr Zuzuarregui) in order to improve confidence that all classified awakenings are true awakenings (rather than misclassifications of REM for example). Through improved classification and better temporal alignment - this has strengthened our results, including an earlier significant change in neurophysiology (subcortical beta) before awakenings (-7.5s).

We have discussed this briefly in the Results:

Polysomnography (PSG) data were collected from the Dreem2 PSG headband which provided automated sleep scoring for 30s epochs according to standard sleep staging (Wake, NREM: N3, N2, N1 and REM) via automated EEG-based sleep staging algorithm which has previously been validated on healthy participants (Fig. 1C)^{26,27}. As N1 generally is difficult to detect and physiologically distinct, we focused our analysis on N2 and N3 stages for NREM sleep (denoted as N2/N3 NREM). Finally, N2/N3 NREM to wakefulness captured by polysomnography were manually re-scored by a board certified sleep physician to corroborate the automated scoring and obtain more precise awakening estimates.

We have also noted this in our Methods section during the description of PSG acquisition:

Finally, to overcome any loss of accuracy incurred through automated sleep staging and to obtain a more precise timing estimate of awakenings (automated sleep staging works on a 30s epoch window), we manually re-scored the N2/N3 NREM to wake after sleep transitions through evaluation of the portable PSG recordings, according to American Academy of Sleep PSG staging rules. Manual scoring was taken as primary where there was disagreement between manual and automated scores.

We have also discussed these points as limitations in the discussion:

To mitigate any limitations of the automated sleep-scoring algorithm, we manually re-scored all awakenings by a board certified sleep physician.

2. The authors note that cortical beta power was higher in PD than in the dystonia patients. This is interesting, because the question whether cortical beta is exaggerated in PD remains a topic of debate. I would like to invite the authors to discuss this in a bit more detail, comparing wake, REM and NREM in this cohort.

This is an excellent suggestion. As mentioned earlier we plan a detailed reporting of REM in a future study. However, following this suggestion we have extended our original analysis and now discussed cortical and subcortical beta power during wake and NREM in PD versus our dystonia control with more details in the results section:

A direct comparison of PD (n=4) vs Dystonia (n=1) revealed that subcortical beta power was lower in the dystonia than all four of the PD participants during N2/N3 NREM sleep (LME model for PD vs Dystonia fixed effect: $\beta = 0.19$; 95%CI= [0.11, 0.28]; p-value = 2.3e-5; n=53). A similar comparison of cortical beta did not show any statistically significant changes at the group level during N2/N3 NREM sleep (p-value=0.34). However, during wakefulness both cortical and subcortical beta was higher in PD compared to dystonia (LME model for PD vs Dystonia fixed effect; cortical beta: $\beta = 0.14$; 95%CI= [0.015, 0.26]; p-value =0.028; subcortical beta: $\beta = 0.36$; 95%CI= [0.31, 0.41]; p-value =2.4e-19).

3. Given the importance of it, I would suggest to add brief mention how extracranial and intracranial signals were synced and how artifacts were removed in the first paragraph of the results text, as this is critical to judge the validity of the subsequent sections for the reader.

This is an important omission in the original text that we remedy here. We have now added the following to the first paragraph of our Results section:

Intracranial and extracranial recordings were synchronized by resampling the extracranial (Dreem headband) data and determining the delays (lag) between the intracranial and extracranial signals by the application of cross-correlation to the accelerometry data from both sources (Fig. 1E; Supplementary Fig. 3). As a secondary validation, included in the protocol were pre-planned synchronized perturbations (x5 taps) of the accelerometers on both the RC+S and the wearable PSG (this secondary validation was performed independently and bilaterally for the two hemispheres) and the final synchronization outcomes were manually inspected for each night. Large artifactual spikes in the subcortical intracranial data were detected by first smoothing the absolute squared data with Gaussian kernel and then finding time periods that exceeded a threshold (determined for each night; see Methods for details). These were removed along with the corresponding cortical data (Supplementary Fig. 3). Finally, the ECG artifacts in the subcortical data were removed using an optimized combination of two ECG data removal algorithms^{28,29}

(Supplementary Fig. 3; see Methods for details) resulting in interpretable cortical and subcortical recordings, even in the presence of DBS.

4. Similarly, describe that sleep stages were defined based on Dreem before the first mention of the results with respect to sleep stages. Consider a half sentence on performance / validation there.

Thank you for noting this. We have now added a discussion on the sleep staging process in the Result section:

Polysomnography (PSG) data were collected from the Dreem2 PSG headband which provided automated sleep scoring for 30s epochs according to standard sleep staging (Wake, NREM: N3, N2, N1 and REM) via automated EEG-based sleep staging algorithm which has previously been validated on healthy participants (Fig. 1C)^{26,27}. As N1 generally is difficult to detect and physiologically distinct, we focused our analysis on N2 and N3 stages for NREM sleep (denoted as N2/N3 NREM). Finally, N2/N3 NREM to wakefulness captured by polysomnography were manually re-scored by a board certified sleep physician to corroborate the automated scoring and obtain more precise awakening estimates.

5. Given the fact that subcortical beta activity is a major hallmark of PD and that non-human primate work that in part inspired this study have mostly focused on beta activity, it was sometimes confusing that the authors chose to highlight changes in delta first, especially in the early results section. I would suggest to consider reporting subcortical and cortical beta first as the hypotheses and expected results can be more easily followed by the reader and therefore may improve readability and match expectations better.

We thank the reviewer for this insightful suggestion. We agree. We have now reorganized our results section and first highlighted cortical and subcortical beta in the results section during our description of spectral power changes, during our reporting of spectral coherence changes, and also in our description of spontaneous awakenings.

6. From my perspective the discussion can be improved. I personally believe that hinting to limitations of previous studies as done in paragraphs 2-4 of discussion is better shortened and shifted to introduction to be rephrased as unmet needs. I got bored reading what others haven't done instead of learning how the authors conceptualize their findings. I guess this is rather an artifact of editorial strategies, as journals increasingly prioritize reviewers comments or ratings with respect to novelty, but often fail to find reviewers who can actually judge the novelty sufficiently. Because of this increasing trend it is understandable that authors devote a large section of text to highlighting the novelty. Here is an expert reviewer who can and does identify this study as clearly novel, so maybe this paragraph just isn't for me. In case a revision is invited, perhaps this section can be replaced by a thorough discussion of the impact of the present study on the current ongoing development of closed-loop neurostimulation therapies for PD and other brain disorders. Specifically, it would be important to discuss the implications for the ongoing

ADAPT-PD trial on beta based adaptive DBS. Furthermore, it should be discussed to what extent cortical signals will be a key for successful closed-loop therapies in PD. I also found the finding that subcortical beta precedes cortical delta very interesting and deserving of more detailed discussion.

It is encouraging to get this kind of feedback regarding the novelty of the study and also support to rewrite as suggested. We have now significantly edited this section and either removed or moved 3 of these paragraphs from discussion to introduction. The remaining paragraph was modified as following:

It is established that during the daytime, subcortical beta oscillations are excessive in PD and potentially contribute to circuit disruption and motor symptoms^{32,33}. Here, we show that subcortical beta oscillations also disrupt cortical slow oscillations during N2/N3 NREM sleep in humans with PD and are partially responsible for awakenings during the night, validating findings from PD models in primates¹⁷. The rise of subcortical beta at least 7.5s before awakenings, at which time delta didn't yet show a statistically significant change, plus the leading of subcortical beta (9s-4s; 3 out of 4 PD participants; Fig. 3D) compared to cortical delta within N2/N3 NREM, support a potential causal relationship between awakenings and subcortical beta in PD. Sleep related cortico-basal network beta fluctuations also have translational implications for emerging adaptive DBS therapies that often use beta as a control signal input. The overall reduction in beta as shown in our subjects would result in a down titration of stimulation in beta triggered aDBS algorithms which could be problematic if higher stimulation amplitudes are beneficial for sleep in a given patient. Conversely, the rapid rise of the subcortical beta prior to awakenings should, in theory, trigger an increase in stimulation amplitude that could be beneficial. Although, the response to beta amplitude would be dependent on the parameterization of the control algorithm and the rapidity of stimulation amplitude changes, with opportunity for personalization. Also, the subcortical beta signal, although captured by the statistical (LME) models, was also partially obscured by a lower SNR in some patients, which potentially poses a challenge for ML algorithms to track and utilize this biomarker for online pre-wake prediction. Further, we show that DBS stimulation, while known to reduce subcortical beta oscillations during wakefulness³⁴, also significantly impacts cortical delta and low beta power during N2/N3 NREM sleep. This finding aligns with previous studies where an increased accumulation of EEG delta power during NREM sleep was found as a result of subthalamic DBS in PD³⁵, but here provides a candidate causal mechanism. Data in our study indicates that DBS therapy appears to improve sleep in PD, at least in part, through direct modulation of beta and delta oscillations.

We are currently addressing the question of whether subcortical recordings are sufficient for sleep classification (compared to cortex) in a data driven ML analysis / report that we plan to present in the coming months. From our analysis so far, a decent summary seems to be that for simpler linear models in the presence of stimulation for a 5 class problem (wake, n1,n2,n3,REM) cortical LFP classifiers significantly outstrip subcortical LFP classifiers. This gap can be narrowed

somewhat by using large amounts of data and more sophisticated models (e.g. deep neural nets) as expected and we hope to share this also early next year.

7. The authors state that the challenge to record invasive neurophysiology is “solved” by sensing enabled devices. I understand that this is strictly speaking true, but it comes with a more general notion that it is no longer difficult to obtain naturalistic data, which is absolutely not true. Therefore, I feel that this sentiment is a massive overstatement that is potentially harmful to future studies that overcome the difficulties of gaining access, overcoming regulatory burdens, developing computational tools and identifying patients who are willing to participate in research with such devices. Recording invasive neurophysiology in naturalistic settings will be a significant challenge for at least another five years. In fact, the authors will very likely not be able to perform this kind of research in five years, because the device and the ECoG strip electrode is no longer supported by manufacturers with currently no real viable alternative in sight.

We thank the reviewer for noting this point and agree with the importance of highlighting the ongoing need for clinical research tools and devices that can perform this kind of research into the future. As such we have now added the following in the discussion highlighting this issue:

One of the major technical aspects of our study was recording full spectrum, time-domain, intracranial cortical and subcortical neural activity during sleep, over multiple nights (n=58). This research leveraged the availability of a critical investigational DBS device (Summit RC+S), which is no longer available. This study data and that by other teams in our research community supports the need for such ongoing DBS research tools to be made available through collaboration between academics, regulators and industry³⁸.

8. For future submissions, please do the reviewers the favor of adding figures directly in text, even if the journals don't want that. It makes reviewing so much more joyful.

We certainly aim to make reviewing joyful! This is a valuable suggestion and we have implemented this in our current revision and will do so for future submissions.

9. The “FP” abbreviation makes it unnecessarily difficult to read, I would suggest to get rid of it. Use activity, oscillations or any other generic term that is readable.

We have now removed the abbreviation from our main text of the manuscript and replaced it with “neural activity”.

10. I would suggest to add a table for the machine learning performances in main text, as the results get to be a bit convoluted.

We have now added a summary table of the machine learning performances in the main manuscript which also now highlights the improved classification results following the manual re-scoring of key epochs:

Table 2: N2/N3 NREM vs wakefulness classification

	Cortex				Subcortex (STN/GPi)			
Epoch	30s		5s		30s		5s	
CV	5-fold	2-fold	5-fold	2-fold	5-fold	2-fold	5-fold	2-fold
Accuracy	93.60±1.40	93.65±1.37	90.60±1.63	90.93±1.61	87.43±3.04	87.15±3.09	80.62±4.08	81.80±3.62
AUC	97.50±0.67	97.53±0.69	95.42±1.10	95.70±1.08	94.02±1.83	93.93±1.85	88.30±3.37	89.23±2.97
Sensitivity	93.88±1.67	93.50±1.14	93.62±1.23	90.70±1.47	86.50±3.26	88.00±2.93	88.72±1.36	82.48±4.66
Specificity	93.30±1.15	93.88±1.60	87.57±2.06	91.13±1.79	88.35±2.85	86.33±3.25	72.53±8.14	81.08±2.86
PPV	93.35±1.19	93.85±1.58	88.32±1.87	91.13±1.74	88.07±2.93	86.55±3.17	77.38±5.13	81.18±3.11
NPV	93.88±1.63	93.48±1.17	93.20±1.38	90.73±1.48	86.80±3.17	87.75±3.02	86.22±1.70	82.50±4.16

Individual machine-learning model performance for N2/N3 NREM vs wakefulness binary classification using cortical data. PPV = positive predictive value, NPV = negative predictive value, U-test = Wilcoxon rank sum test, AUC = Area under the receiver operating characteristic curve, CV= Cross-validation.

We have also simplified the corresponding results section:

Our results showed strong classification performance both using sensorimotor cortex and subcortical regions (Cortex: 93.6±1.4% accuracy; STN/GPi: 87.4±3% accuracy; Fig. 5B; Table 2; Supplementary Fig. 5A; Supplementary Table 2; Supplementary Table 3) for N2/N3 NREM vs wake classification across all PD participants at the standard 30s epoch level. We then probed N2/N3 NREM vs wake classification at a finer temporal scale (reducing from 30s to 5s epoch) for rapidly dynamic awakening detection- towards sleep-specific aDBS development. Specifically, we trained participant-specific SVM models using 5s epochs achieving high classification performances (Cortex: 90.6±1.6% accuracy; STN/GPi: 80.6±4.1% accuracy; Fig. 5C; Table 2; Supplementary Fig. 5B; Supplementary Table 2; Supplementary Table 3) for N2/N3 NREM vs wakefulness.

11. Fig. 2 D subcortex one signal looks (PD3 GPi) odd, did the authors look at the change in wakefulness to rule out stimulation induced artifacts?

Yes - we agree that in this patient the SNR of the subcortical data was lower and partially affected by artifacts. We did use a number of techniques including spike detection, anti-aliasing and bandpass filtering but were unable to recover more signal than shown for this one night of data. The subcortical data on the single ON stimulation night (PD3 GPi) during the 1 night ON and 1 night OFF data collection period were of low quality (flatter 1/f characteristics with minimal band-specific activity) compared to other nights. The data had low SNR in all frequencies with some noisy peaks. Note that we didn't report any statistically significant neurophysiology that involves

this data (ON vs OFF in subcortical activities) but we have shown this data in the plot for completeness. Nonetheless, we have now noted this low SNR issue in the results:

No significant changes in subcortical activities during N2/N3 NREM were observed in ON vs OFF power spectrum comparisons, however, changes in subcortical baseline power levels in the ON versus OFF DBS state and lower SNR (signal-to-noise ratio) in data from GPi of PD3 (ON state) may have obscured any underlying changes.

12. In Figure 3 B OFF Figure 4 box plots individual data points are missing, please add or change plot type.

While we had many nights per subject of recordings ON stimulation, we only had one OFF night per subject as this was uncomfortable for patients. For clarity therefore, we have changed the plot type and added descriptions to the figure legend to mitigate the confusion:

(B) Average Spearman's rho correlation between subcortical beta power and cortical delta power for all 4 PD participants across multiple nights in ON (left) and in OFF (right) stimulation during N2/N3 NREM. Each bar shows average correlation for one participant and each point shows correlation across one night with data pooled from both hemispheres. Stem plot in OFF stimulation shows correlation across the single OFF night per participant.

13. Add spearman's rho values and p-values to scatter plots in Figure 3 as the massive point clouds are hard to interpret.

We agree. As each panel in the scatter plots of Figure 3 shows a single night of data from both hemispheres in that individual patient, we elected to show results of linear mixed effect models (LME) for that night that could account for the the two hemispheres as random effects instead of correlations. We have now updated Figure 3 with linear fit to the data for better visualization and included the single night statistics in the legend:

C. Cortical-subcortical delta-beta interaction

E. Cortical delta-beta interaction

(C) Scatter plots depicting the correlation between subcortical beta (13-31 Hz) power and cortical delta (1-4 Hz) power during N2/N3 NREM sleep in 4 PD participants during ON stimulation; STN (*brown and red*), and GPI (*blue, light blue*). Each point represents data from one 5s N2/N3 NREM sleep epoch. Each plot is data from one night pooled from both hemispheres for one participant. LME models were constructed for the single night data

shown here of each panel for cortical delta with subcortical beta as fixed while accounting for the random effect of hemisphere side (PD3: $\beta = -0.58$, p-value $< 1e-300$; PD9: $\beta = -0.87$, p-value = $1.2e-144$; PD2: $\beta = -0.41$, p-value = $1.8e-129$; PD7: $\beta = -0.41$, p-value = $1.3e-139$). (E) Interactions between cortical delta and cortical beta activities, examined as a control for cortical delta-subcortical beta during N2/N3 NREM. The bar plot (*left*) shows average Spearman's rho correlation between cortical delta and beta power for all 4 PD participants across multiple nights, ON stimulation. Each bar shows average correlation for one participant and each point shows correlation across one night with data pooled from both hemispheres. The scatter plots (*middle and right*) show cortical delta and beta power in 2 PD participants during ON stimulation for two representative PD participants for a single night. Each point represents data from one 5s N2/N3 NREM sleep epoch. LME models were constructed for the single night data of each panel similar to panel C shown here for 2 participants (PD2: $\beta = 0.18$, p-value = $4.7e-12$; PD9: $\beta = -0.36$, p-value = $1.7e-60$).

For further clarification, we also included new results for LME model of each PD participant individually (previous LME model pooled all subjects) across all nights with ON stimulation (in contrast to the single night panels in the figure) in the Results, first for the subcortical beta and cortical delta interaction which demonstrated that this relationship was significant for each subject at the within subject level:

LME models for each PD participant individually showed a negative fixed effect of subcortical beta on cortical delta during N2/N3 NREM sleep in ON stimulation, across all nights (PD3: $\beta = -0.43$, p-value = $1.24e-19$; PD9: $\beta = -0.59$, p-value = $7.1e-39$; PD2: $\beta = -0.47$, p-value = $1.49e-54$; PD7: $\beta = -0.55$, p-value = $1.48e-13$).

and later comparing cortical beta and delta which was significant in all 4 subjects - but showed a positive correlation in one subject):

LME model for 3 out of 4 PD participants showed negative fixed effects of cortical beta on cortical delta during N2/N3 NREM sleep in ON stimulation (PD3: $\beta = -0.67$, p-value = $1.72e-54$; PD9: $\beta = -0.36$, p-value = $1.9e-9$; PD2: $\beta = 0.23$, p-value = $9.4e-11$; PD7: $\beta = -0.43$, p-value = $2.3e-20$).

Note - that our previously included analysis demonstrated that the subcortical - cortical, beta - delta interaction was stronger than the single site cortical - subcortical interaction.

14. Add description/legend to black line to Figure 4 inside panels A-D

We have updated Figure 4 with a legend to the black line to label this "Accelerometry movement":

Reviewer #2 (Remarks to the Author):

The paper of Anjum et al. explores intriguing aspects of subcortical and cortical oscillations during sleep, shedding light on major scientific findings and potential clinical applications. There significant discovery is the observed correlation between delta and beta connectivity during sleep, where delta increased and beta decreased, showcasing enhanced coherence under deep brain stimulation.

We thank the reviewer for highlighting the significant discovery of delta-beta interaction during sleep and with DBS.

The clinical translation of this finding is noteworthy, particularly the ability to predict NREM N2/3 sleep versus wakefulness within a short 5-second window.

We agree with the reviewer and have added more text regarding translational implications (also requested by Reviewer 1) including:

Sleep related cortico-basal network beta fluctuations also have translational implications for emerging adaptive DBS therapies that often use beta as a control signal input. The overall reduction in beta as shown in our subjects would result in a down titration of stimulation in beta triggered aDBS algorithms which could be problematic if higher stimulation amplitudes are beneficial for sleep in a given patient. Conversely, the rapid rise of the subcortical beta prior to awakenings should, in theory, trigger an increase in stimulation amplitude that could be beneficial. Although, the response to beta amplitude would be dependent on the parameterization of the control algorithm and the rapidity of stimulation amplitude changes, with opportunity for personalization. Also, the subcortical beta signal, although captured by the statistical (LME) models, was also partially obscured by a lower SNR in some patients, which potentially poses a challenge for ML algorithms to track and utilize this biomarker for online pre-wake prediction.

While the paper holds promise for the broad readership of the journal, several critical points require attention, including the need for direct testing of translational capabilities and addressing methodological concerns.

Major points of concern:

i. The impact of medication withdrawal ("Med off") on delta-beta coherence raises questions about the differential effects of dopamine and DBS. A detailed exploration of how DBS acts similarly or differently compared to dopaminergic interventions is crucial for a comprehensive understanding.

We thank the reviewer for the suggestion. Although we agree that it would be interesting to formally investigate this, we note that this is potentially challenging to obtain multi-night sleep data from participants OFF medications, due to the discomfort and potential clinical instability from multiple nights of dopamine withdrawal. This is compounded by the prolonged pharmacokinetics of many dopamine medications which would therefore require additional prior daytime withdrawal

of medications. Note is also made here of the central finding of the study that during sleep (even in the ON medication state), subcortical beta appears to be increased, interferes with delta waves and causes awakening. Given the strong established relationship between exogenous dopamine administration and subcortical beta suppression - we would expect the effects we found to be even stronger if dopamine was restricted, however this could be formally tested in a future study.

However, despite the challenges of collecting this data, we do appreciate the suggestion of exploring the neurophysiological changes resultant from dopaminergic modulations. We have therefore sought to address this indirectly by dividing each night into four quadrants and determining whether the effects are stronger at the end of the night (4th quadrant) when the sinemet has worn off compared to the beginning of the night (1st quadrant). We utilized data from 1st and 4th quadrant of all 4 PD participants during ON stimulation and utilized a linear mixed effect (LME) model of cortical delta with subcortical beta and quadrant (1st or 4th) as fixed effects. The LME model showed a statistically significant fixed effect of quadrant (beginning/end of the night) on the cortical delta ($\beta = -0.005$, 95%CI= [-0.008, -0.002], p-value = 0.001, n=99,155) while subcortical beta showed statistically significant negative effect as expected ($\beta = -0.46$, 95%CI= [-0.5, -0.41], p-value = 2e-98). We have now added this into the manuscript and thank the reviewer for this suggestion:

Finally, we investigated the impact of dopaminergic medications by dividing each night into four quadrants and measuring effects in the neurophysiology at the end of the night (when the dopaminergic medications had partially worn off) compared to the beginning of the night. We utilized data from 1st and 4th quadrant of all 4 PD participants during ON stimulation and implemented a linear mixed effect (LME) model of cortical delta with subcortical beta plus time quadrant (1st or 4th) as fixed effects. The LME model showed a statistically significant fixed effect of quadrant (beginning/end of the night) on the cortical delta ($\beta = -0.01$, 95%CI= [-0.02, -0.006], p-value = 1.7e-5, n=99,155) while subcortical beta showed a statistically significant negative effect on cortical delta as expected while accounting for both quadrants ($\beta = -0.44$, 95%CI= [-0.49, -0.4], p-value = 3.2e-87). The negative effect of subcortical beta was significantly stronger during the 4th quadrant (effect of 1st/4th quadrant on the interaction: $\beta = -0.006$, 95%CI= [-0.01, -0.002], p-value = 0.003).

ii. The limited sample size of only four subjects is a notable concern. A power analysis is essential to evaluate the stability of findings across subjects. Additionally, distinctions between the effects of DBS on two different target areas (2 GPi and 2 STN) should be explored, considering potential variations in response.

We agree that performing an across subject analysis (e.g. single night of data - correlated to single clinical outcome measure) with 4 patients would likely be under powered. However, we here take a highly powered within-subject approach and primarily investigate within-subject hypotheses from a previous primate study (note: the primate study on which this study was based was completed in only 2 animals). Having the extensive (hundreds of hours) of within-subject

recordings as afforded by our technical setup allows us to perform analyses that are normally restricted to within-subject primate work. We address this in the limitations section:

We also report many nights of recordings per participant (n=57 total), but from a relatively small number of subjects, which supported highly statistically powered LME analyzes that modeled within, as well as across, participant effects - similar to the strengths of primate research. This approach was felt to be well suited to looking for the within-subject cortico-subcortical interactions which were the primary focus of this study. However, evaluation of across subject factors, including analysis of how beta - delta interactions predict clinical outcomes at scale would require a different study design that included large numbers of subjects (but would also require much less within-subject data).

However, in light of the concerns by the reviewer, we have now also added the per subject statistics (in addition to the group level LME model results previously reported) to highlight that these effects were individually (highly) significant for each subject and therefore, not surprisingly - also significant at the group level when combined using an LME model:

LME models for each PD participant individually showed a negative fixed effect of subcortical beta on cortical delta during N2/N3 NREM sleep in ON stimulation, across all nights (PD3: $\beta = -0.43$, p-value = $1.24e-19$; PD9: $\beta = -0.59$, p-value = $7.1e-39$; PD2: $\beta = -0.47$, p-value = $1.49e-54$; PD7: $\beta = -0.55$, p-value = $1.48e-13$).

Second, in order to provide confidence in this approach in humans, we have now conducted a power analysis to evaluate the stability of our LME models which confirmed a strong statistical power of 0.97. Specifically, to evaluate the robustness of the statistical findings across participants, we conducted a power analysis where we simulated data from the LME model of cortical delta with subcortical beta (accounting for the dependency between left and right hemispheres) and multiple nights within each participant. We used values of the means and standard deviations from multi-night data (n=53) from 5 participants during ON stimulation. Specifically, we generated subcortical beta from a normal distribution with mean -1.0895 (log value therefore negative) and standard deviation 0.2487 and generated the random error from a mean zero normal distribution with standard deviation 0.7484 (four times the value estimated from data). In addition, we generated the random effects from a centered normal distribution with standard deviation 0.9 for night and participants and 0.02 for hemisphere. Finally, we generated cortical delta with mean -0.5 (log value therefore negative) along with fixed and random effects. We simulated 10,000 datasets under null and alternative hypotheses, and we rejected the null hypothesis if the p-value of the estimated effect was less than 0.05. The simulation results demonstrated that with 5 subjects and 11 nights per subject, we will achieve a power value 0.97 (beta = 0.03) to test the main hypothesis.

Investigating the possible variations between the effects of DBS on GPi and STN is an excellent suggestion. The figures indeed suggest a potential variation due to DBS in NREM. We have now added target area as a fixed effect in our LME models which showed a statistically significant effect of target area in the modulation of cortical sigma power and cortico-basal sigma coherence

with DBS on versus OFF. We have now added the modulation of cortical sigma power in the Results during the spectral power changes in NREM :

Location of the DBS leads in the subcortical structure (GPi/STN) did not show any statistically significant effect in the changes of cortical delta (p-value= 0.06) and alpha (p-value= 0.4) but had a statistically fixed effect only in decreased sigma power (13-15 Hz; β = -0.56, 95%CI= [-1.02, -0.1], p-value = 0.026) in GPi compared to STN.

and added the modulation of cortico-basal sigma coherence in the Results during the changes in functional connectivity in NREM :

In our ON vs OFF DBS analysis (PD participants only; n=4; Table 1), we also noted a statistically significant further decrease in cortico-basal sigma (13 - 15 Hz) coherence during ON stimulation compared to OFF (LME model with ON/OFF condition as fixed and participants as random effects; β = -0.014, 95%CI= [-0.021, -0.006], p-value = 0.005) during N2/N3 NREM with a statistically fixed effect of the location of DBS leads (STN/GPi: β = 0.08, 95%CI= [0.002, 0.16], p-value = 0.047).

iii. Relying on a single dystonia patient as a control may not provide sufficient grounds for drawing meaningful conclusions. A more diverse control group is needed to strengthen the validity of the findings and draw accurate comparisons. As an alternative to dystonia patients, a dataset from patients with epilepsy undergoing stereo-electroencephalography (SEEG) could be considered.

In this study we primarily aimed to test in patients with Parkinson's disease the hypothesis that subcortical beta negatively interacts with cortical beta, motivated by a previous primate study (2 primates) that had demonstrated this in non-human models of PD. Due to the unique nature of the recording infrastructure and equipment - namely a high resolution sensing enabled pacemaker, chronic electrocorticography and chronic subcortical electrodes we were limited to working with the patients enrolled in our parent trial. Further, a key part of the investigation here centered on the pathophysiological cause of the large number of awakenings seen, which is relatively specific to PD. As highlighted by Reviewer 1, unfortunately the RC+S device is no longer in production and therefore we do not have the option of enrolling further patients to expand this cohort. We have now included in the acknowledgement section that this is a "comparison" patient rather than a formal "control":

Finally, our comparison participant was a single cervical dystonia patient (rather than a formal control group) reflecting the uniqueness of this participant cohort, with high-resolution sensing-enabled pulse generators and chronically implanted ECoG electrodes.

We also now state that this is exploratory evidence which motivates future larger studies:

However, despite this, and in view of the large within-participant dataset size and linear mixed modeling, we were able to find exploratory evidence in support of a difference

between the dystonia participant and the PD group, which motivates future larger studies with formal comparison at the group level.

We appreciate the reviewer for recording an alternative multi-subject sEEG dataset. However, this option is limited by the fact that sEEG epilepsy patients are not using implanted with paired ECOG and STN or GPi electrodes, which would limit comparison, in addition to the practical challenges of repeating the study and data collection in another separate cohort.

Finally, we would like to note that although we reported some effects of PD vs comparison Dystonia in neurophysiology, we kept the claims to be minimal and focused predominantly on the within PD only results. This can be seen in our machine learning models, effects of DBS, changes in neurophysiology during awakenings etc. .

iv. The predictive performance of the N2/3 NREM sleep prediction (5-second recording) holds translational value for adaptive applications. However, to establish robust predictive capabilities, it is imperative to demonstrate performance across an out-of-sample cohort comprising a minimum of 15-20 subjects. This would enhance the generalizability and reliability of the proposed prediction model.

We thank the reviewer for noting the importance of our classification results. It is worth noting here that we built individual, personalized ML models, rather than a general model (that is designed to work on any subject) and so are not here seeking to demonstrate or validate a generalizable model. i.e. our results were for patient-specific machine-learning models which would not have affected by adding more participants.

Additionally, obtaining overnight recordings as suggested from an additional 15-20 subjects with 10 nights per subject would amount to 200 further nights of data recording. This is currently infeasible given the available cohort (see above), discontinuation of the RC+S device and available research resources in the laboratory within the time frame of this study and project as it took us 2 years to collect the current data. Finally, the classification of N2/N3 NREM vs wakefulness was not the main objective of this study. Rather, the primary purpose of was to test the hypothesis regarding beta-delta interactions and sleep disruption and the ML was provided as a secondary analytic method to show support for that hypothesis (from the conventional LME statistics) as well as to show translational implications. A full ML based characterization of sleep is another extensive piece of work (including the 5 class problem) that will be reported next year (see answer to Reviewer 1).

Nevertheless, in order to address the reviewers' concerns regarding ML model validity, we have now completely re-run the ML analyzes again , but now adding significantly more stringent validation to increase the confidence in the ML models. First, we have now included results from a new 2-fold cross-validation where only 50% of the data were used for training (significantly reducing the training data set and increasing the testing set) and the remaining data were tested. These results show that the performance was consistent across cross-validation schemes. We have added these results in the Results section:

Table 2: N2/N3 NREM vs wakefulness classification

	cortex				Subcortex (STN/GPi)			
Epoch	30s		5s		30s		5s	
CV	5-fold	2-fold	5-fold	2-fold	5-fold	2-fold	5-fold	2-fold
Accuracy	93.60±1.40	93.65±1.37	90.60±1.63	90.93±1.61	87.43±3.04	87.15±3.09	80.62±4.08	81.80±3.62
AUC	97.50±0.67	97.53±0.69	95.42±1.10	95.70±1.08	94.02±1.83	93.93±1.85	88.30±3.37	89.23±2.97
Sensitivity	93.88±1.67	93.50±1.14	93.62±1.23	90.70±1.47	86.50±3.26	88.00±2.93	88.72±1.36	82.48±4.66
Specificity	93.30±1.15	93.88±1.60	87.57±2.06	91.13±1.79	88.35±2.85	86.33±3.25	72.53±8.14	81.08±2.86
PPV	93.35±1.19	93.85±1.58	88.32±1.87	91.13±1.74	88.07±2.93	86.55±3.17	77.38±5.13	81.18±3.11
NPV	93.88±1.63	93.48±1.17	93.20±1.38	90.73±1.48	86.80±3.17	87.75±3.02	86.22±1.70	82.50±4.16

Individual machine-learning model performance for N2/N3 NREM vs wakefulness binary classification using cortical data. PPV = positive predictive value, NPV = negative predictive value, U-test = Wilcoxon rank sum test, AUC = Area under the receiver operating characteristic curve, CV= Cross-validation.

And discussed this in the Methods:

In addition to 5-fold cross-validation, we also implemented a 2-fold (50% of the data were used for training and the remaining data were tested) cross-validation scheme to investigate the robustness of our performance.

And finally noted the consistency in the Results:

Finally, our participant-specific ML models showed consistent performance across the cross-validation schemes (2-fold vs 5-fold; Table 2).

Secondly, we have now also performed subject-wise cross-validation (test the performance on each PD participant after training the model using the remaining 3 participants; repeated for all 4 PD participants). We found that even at the subject-wise fold, the machine learning model was able to achieve more than 80% accuracy for cortical features (despite these being primarily designed as individualized / personalized models). The 10% drop in the performance of cortical features illustrates the advantage of patient-specific ML models. The performance for subcortical features dropped to ~75% accuracy, illustrating the effect of the location of subcortical leads (GPi/STN) in the across-subject ML models.

N2/N3 NREM vs wakefulness classification across-subjects:

Epoch	cortex		Subcortex (STN/GPi)	
	30s	5s	30s	5s
Accuracy	82.68±4.63	81.08±3.68	73.70±7.96	74.33±7.92
AUC	90.10±3.88	89.50±1.93	86.48±3.52	82.60±3.94
Sensitivity	81.63±5.69	81.43±6.94	70.83±10.24	75.13±11.00
Specificity	87.25±7.85	83.25±11.37	85.83±5.17	75.08±7.03
PPV	95.03±3.37	94.18±3.87	94.35±2.00	90.88±2.67
NPV	61.28±5.88	61.93±8.76	51.23±10.00	53.65±10.28

Table description: Individual machine-learning model performance for N2/N3 NREM vs wakefulness binary classification using cortical data. PPV = positive predictive value, NPV = negative predictive value, U-test = Wilcoxon rank sum test, AUC = Area under the receiver operating characteristic curve, Data from all 4 PD participants during ON stimulation. Performance was measured by subject-wise cross-validation (average performance on each subject while being trained on the remaining 3 subjects).

Finally, in addition to extra strict cross validation, across subject generalizability testing we now have implemented other unconstrained and powerful machine-learning models (deep neural networks) to illustrate the model invariance of our results. Unlike the SVM approach where we manually provided specific bandpowers as features, these new deep neural networks utilized raw time series data as inputs and performed the classification by generating their own features. In particular, we utilized convolutional neural networks (CNN) and residual network (ResNet) architecture with Gated recurrent units (GRU) and multi-head attention Transformer layers. Cortical data with 5s epochs were utilized for this purpose. Unlike the SVM approach where we manually provided specific bandpowers as features, these deep neural networks utilized raw time series data as inputs and performed the classification by generating their own features. Despite the major data differences compared to the SVM approach (e.g., time series vs band power input, lack of feature selection) and the training (e.g., partial hyperparameter and network architecture optimization), these new models provided similar performance to our SVM approach showing the robustness of our ML results. We have now added these below:

N2/N3 NREM vs wakefulness classification with different approaches

method	Network details	CV	Input data	ACC	AUC	SEN	SPEC
SVM	RBF kernel	5-fold	Bandpower	90.60±1.63	95.42±1.10	93.62±1.23	87.57±2.06
SVM	RBF kernel	2-fold	Bandpower	90.93±1.61	95.70±1.08	90.70±1.47	91.13±1.79
CNN	Layer:16-16-32 FC:16-8-1	5-fold	Time series	91.08±1.54	96.49±0.92	92.01±2.05	90.15±1.23
ResNet	Layer:5x64 FC:16-8-1	5-fold	Time series	91.32±1.61	96.62±0.99	91.38±1.81	91.25±1.62
ResNet+ GRU	Layer:5x64 GRU:16-1	5-fold	Time series	91.38±1.65	96.73±0.95	91.48±2.15	91.27±1.19
ResNet+ BiGRU	Layer:6x64 BiGRU:16-1	5-fold	Time series	91.32±1.69	96.74±0.93	90.77±2.83	91.87±0.93
ResNet+ Transformer	Layer:6x64 Tx: 32 H:2	2-fold	Time series	90.59±1.57	96.19±0.94	88.82±2.20	92.35±0.96

Table description: Individual machine-learning model performance for N2/N3 NREM vs wakefulness binary classification using cortical data in 5s epochs. ACC= accuracy, SEN=sensitivity, SPEC=specificity, AUC = Area under the receiver operating characteristic curve. 5-fold cross-validation was applied. GRU= Gated Recurrent Unit, BiGRU= Bidirectional GRU, CNN=Convolutional neural network, ResNet=Residual Network, SVM= Support vector machine. RBG= Radial basis gaussian. Data from all 4 PD participants were utilized. Except for the SVM model, no class balance was applied before training. The training of the deep neural networks were done using 64 epochs with batch size of 64 and no class balance was implemented. The deep neural network models were run using Keras under Python 3.8.18 with TensorFlow 2.13.1 in a machine with linux operating system (Ubuntu 22.04.3 LTS). The machine had 80 core Intel(R) Xeon(R) Gold 6242R CPU @ 3.10GHz CPU, 512 GB RAM and Nvidia GA102GL RTX A5000 24GB GPU.

We believe that these analyses and performance comparisons have provided strong support for the robustness of our models to reduced training dataset size, model invariance to ML model type and also a suggestion of the generalizability and robustness of our reported performance.

We plan a further follow up methodological paper on ML classification that comprehensively addresses these questions.

v. The 'perceive' library's ECG removal technique using QRS interpolation, contrary to the method described in the Hammer et al. 2022 paper, fails to effectively eliminate T waves in the ECG signal, leaving behind residual low-frequency artifacts. Even employing alternate methods such

as SVD or template subtraction leads to the inadvertent removal of low-frequency power, resulting in a reduction of spectral content. Given the relevance of delta bands in this study, the approach to mitigate this discrepancy remains unclear. To gain a comprehensive understanding, it would be beneficial to visually compare the power spectrum pre- and post-ECG removal and should be presented.

We were not 100% clear in our original write up -apologies. We primarily utilized the template subtraction pipeline (PerceptHammer²) proposed and developed at UCSF by Hammer et al. 2022 and the 'Perceive' library³ was used in order to generate an initial template of the ECG artifact for the PerceptHammer pipeline. We have detailed our ECG artifact removal procedure in the Methods:

The subcortical data underwent artifact removal for ECG interference through the application of an optimized combination of two complementary ECG data removal algorithms ('PerceptHammer' and 'Perceive' library; Matlab; Supplementary Fig. 6)^{28,29}. For each 10 minute non-overlapping window of subcortical FP data, the 'Perceive' library²⁸ was first utilized to generate an initial starting seed of ECG artifact template based on the presence of characteristic sharp QRS-like signal deflections. This was done to personalize the initial seed template and account for the variations of ECG artifacts that occur across hemispheres (left/right side), participants and even duration of the night. For each night, all ECG artifact templates for 10 minute windows found by the Perceive method were averaged which was then fed to the Template subtraction pipeline (PerceptHammer²⁹) as an *initial* template seed for the actual ECG artifact detection and removal operation. Notably this second algorithm uses Woody's adaptive filter to identify locations of the artifact and then update the template recursively to improve it further. The template subtraction pipeline was applied separately for each 10 minute non-overlapping window with the same initial template seed. This window-wise ECG removal was implemented to account for any changes of ECG artifact throughout the night. Finally, forced searches were conducted by the PerceptHammer pipeline for artifacts missed by the adaptive filter. In order to avoid the template locking into a low-frequency rhythmic neural activity during the recursive update which results in the removal of low-frequency contents of neural activities, we compare the final template formed by Woody's adaptive filter with the initial template seed generated by the Perceive method via normalized cross-correlation. If the maximum cross-correlation is less than a predetermined threshold of 0.9, the results were rejected and the PerceptHammer pipeline was re-applied via forced searches without any recursive update of the initial template.

We have added a new supplemental figure to illustrate our procedure:

A. Perceive: generating ECG template

B. PerceptHammer: successful ECG removal

C. ECG removal pipeline (Perceive+PerceptHammer)

Supplemental Figure 6: ECG artifact removal procedure. (A) ECG template generation by Perceive method. Top left panel shows the ECG template found in the data, the top right panel shows power spectrum comparison between pre and post ECG removal done by Perceive and the bottom panel shows pre and post ECG removal time series comparison for a single subject. (B) ECG artifact removal by PerceptHammer method with initial template seed provided by Perceive. The inset plot shows the same individual templates as above (multiple colors) final template generated by PerceptHammer during the process through recursive template update. (C) Complete ECG artifact removal pipeline implemented in this study. Perceive method generates initial ECG template seed. PerceptHammer attempts to remove ECG artifacts with default mode. If the final updated

template doesn't sufficiently match with the initial template (maximum normalized cross-correlation < 0.9) then re-run PerceptHammer with recursive template update disabled.

The reviewer posed a valid concern that ECG artifact removal procedure might lead to the inadvertent removal of low-frequency power, resulting in a reduction of spectral content. This is indeed an important factor given the relevance of delta bands in our study. Again, we were not fully clear about our methodology in the manuscript. We primarily utilized the PerceptHammer² pipeline for the ECG artifact detection and removal. Furthermore, utilizing the Perceive library, we implemented a safeguard mechanism to detect and avoid the unwanted removal of low-frequency contents from our FP data. We have now clarified our approach in the Methods:

In order to avoid the template locking into a low-frequency rhythmic neural activity during the recursive update which results in the removal of low-frequency contents of neural activities, we compare the final template formed by Woody's adaptive filter with the initial template seed generated by the Perceive method via normalized cross-correlation. If the maximum cross-correlation is less than a predetermined threshold of 0.9, the results were rejected and the PerceptHammer pipeline was re-applied via forced searches without any recursive update of the initial template.

We have now also compared the power spectrum of pre and post ECG removal procedure as the reviewer advised. The comparison showed minimal loss of the low-frequency spectral contents of ECG removal procedure (see below). Rather, the most distinct spectral changes were in a broad range (10-40 Hz) highlighting the effect of removing ECG artifacts. Note that ECG artifacts have a broad spectral content due to their sharpness in time domain. Furthermore, these changes also replicate the effects of ECG removal reported by Hammer et al. through PerceptHammer². We now have described the effects of the ECG artifact removal especially in the low frequency spectral content in an additional panel in supplemental figure 7:

Supplemental Figure 6: (D) Spectral power density comparison of the original subcortical data over one night from one PD participant (blue; Pre-ECG removal) and cleaned data (red: Post-ECG removal) over full frequency range (left; 0-90 Hz) and over low frequency

range (right; 0-6 Hz) showing minimal loss of the low-frequency spectral contents of ECG removal procedure. Major spectral changes following artifact removal were in a broad range (10-40 Hz). Of note, ECG artifacts have a broad spectral content due to their sharpness in time domain. These changes are consistent with the pre and post ECG removal spectral changes observed by Hammer et al. (2022).

Finally, we would like to make a note that any inadvertent removal of low-frequency power that can affect the delta band will only dilute our effects (leading to a lower SNR and weakening the statistical outcomes) which highlights that our results are conservative and could get better with higher data quality.

vi. Under the 'Power Spectrum Analysis' section, the normalization of power spectra using the total power within the 0-50Hz range is specified. However, in the subsequent 'Beta-Delta Correlation Analysis,' the exclusion of beta power (14-30 Hz) in subcortical beta and cortical delta is puzzling. This omission could potentially skew the distribution and bias the correlation towards the beta band, impacting the accuracy of the results. The rationale behind this exclusion lacks sufficient explanation and may introduce distortions in the findings.

We thank the reviewer for addressing this issue which has not been adequately clarified by us. Our rationale for excluding beta was that wide band normalization of power spectra effectively enforces that the total power of all frequencies be a constant. Therefore, using a wide band normalization scheme - if beta power (13-31 Hz) rises in a data window, and one is using a wide band normalization scheme, normalized power in all other frequencies will be "lowered". Given that we were looking precisely for a negative interaction between beta and delta, we were concerned that including beta within the normalization scheme might inadvertently create a negative correlation between beta and delta through simple mathematics. To avoid this issue and to be conservative in our analyses, we avoided the beta band in the normalization procedure. This methodology had been previously implemented in the non-human primate study⁴ that investigated delta-beta interaction for the first time on primates.

However, for the sake of completeness and transparency, we have computed the delta-beta interaction results with uniform (wide band) normalization which we provide here below:

Delta-beta interaction with uniform normalization procedure by 0-50 Hz:

LME modeling using band powers of N2/N3 NREM epochs from all participants (Cervical dystonia and PD participants; accounting for the dependency between left and right hemispheres and multiple nights within participants; n=241,643) showed an overall negative fixed effect of subcortical beta power on cortical delta power ($\beta = -0.39$, 95%CI: [-0.46, -0.32], p-value = 5.6e-30) during N2/N3 NREM sleep, ON stimulation. Additionally, the LME model revealed a fixed effect of PD vs Dystonia state ($\beta = 0.17$, 95%CI: [0.11, 0.22], p-value = 1e-8), demonstrating that this effect was greater in the PD participants than our dystonia comparison participant. A negative fixed effect of subcortical beta power on cortical delta power was also obtained through an LME model in PD participants during N2/N3 NREM in the OFF stimulation condition ($\beta = -0.43$, 95%CI: [-0.52, -0.33], p-value = 8.6e-19; n=18,226).

We have also recreated the core figures using this wide band normalization:

Inverse relationship between subcortical beta and cortical delta activities during N2/N3 NREM: (A) Example of subcortical beta (purple) and cortical delta (green) power during N2/N3 NREM in a single night from one PD participant (PD3) during ON stimulation depicting the inverse relationship in temporal domain. The delta and beta powers were smoothed with a 20-point gaussian kernel. (B) Average Spearman's rho correlation between subcortical beta power and cortical delta power for all 4 PD participants across multiple nights in ON (left) and in OFF (right) stimulation during N2/N3 NREM. Each bar shows average correlation for one participant and each point shows correlation across one night with data pooled from both hemispheres. Stem plots in OFF stimulation show correlation across the single OFF night per participant. (C) Scatter plots depicting the correlation between subcortical beta (13-31 Hz) power and cortical delta (1-4 Hz) power during N2/N3 NREM sleep in 4 PD participants during ON stimulation; STN (brown and red), and GPi (blue, light blue). Each point represents data from one 5s N2/N3 NREM sleep epoch. Each plot is data from one night pooled from both hemispheres for one participant. LME models were constructed for the single night data of each panels for cortical delta with subcortical beta as fixed while accounting for the random effect of hemisphere side (left/right) and the corresponding coefficient β and p-value are provided.

Comparing the above results and figures with the ones in the main manuscript, it is evident that both normalization schemes reveal similar visual and statistically significant delta-beta interactions during ON stimulation ($\beta = -0.39$ in 0-50 Hz normalization; $\beta = -0.36$ in previously reported normalization) and OFF ($\beta = -0.43$ in 0-50 Hz normalization; $\beta = -0.4$ in previously reported normalization). Also, fixed effects of PD vs Dystonia state were statistically significant

and similar in both cases ($\beta = 0.17$ in 0-50 Hz normalization; $\beta = 0.16$ in previously reported normalization).

Finally, to reduce any potential confusions for the readers, we have added the following in the Methods section:

However, we note that both normalization methods (with and without excluding beta range during normalization) provided highly similar statistical outcomes and conclusions.

Additionally, the use of a log transform in the analysis complicates the interpretation of the resulting spectrum by compressing higher frequency bands more than lower frequency ones, introducing complexity into the interpretation process.

On review, both with and without log transform provided the same conclusions. We providing the delta-beta interaction results without log transform below:

LME modeling using band powers of N2/N3 NREM epochs from all participants (Cervical dystonia and PD participants; accounting for the dependency between left and right hemispheres and multiple nights within participants; n=241,643) showed an overall negative fixed effect of subcortical beta power on cortical delta power ($\beta = -1.1$, 95%CI: [-1.4, -0.9], p-value = 1.4e-24) during N2/N3 NREM sleep, ON stimulation. Additionally, the LME model revealed a fixed effect of PD vs Dystonia state ($\beta = 0.05$, 95%CI: [0.001, 0.1], p-value = 0.04), demonstrating that this effect was greater in the PD participants than our dystonia comparison participant. A negative fixed effect of subcortical beta power on cortical delta power was also obtained through an LME model in PD participants during N2/N3 NREM in the OFF stimulation condition ($\beta = -0.9$, 95%CI: [-1.5, -0.3], p-value = 0.003; n=18,226).

As seen from the above and comparing with our reported results in the main manuscript with log transform, it is evident that the conclusions from statistical analyses remain constant. In particular, both cases provide statistically significant delta-beta interaction during ON stimulation ($p\text{-value} = 1.4e-24$ without log; $p\text{-value} = 2.5e-30$ with log) and OFF ($p\text{-value} = 0.003$ without log; $p\text{-value} = 2.5e-17$ with log). Also, fixed effects of PD vs Dystonia state were statistically significant in both cases ($p\text{-value} = 0.04$ without log; $p\text{-value} = 1.7e-7$ with log). The correlation plots in panel B remain unchanged while panel A showed changes in cortical data that are orders of magnitudes higher than subcortical beta without log transform. To the best of our knowledge, log transform is a well known operation to handle changes in the order of magnitudes, omitting log transform makes their relationship more nonlinear and so statistical models such as LMEs become less robust. We did not find any compression of high frequency due to log transform but rather the high and low frequency activities become more comparable (compression of high amplitude low frequencies). Furthermore, most power spectrums reported in the literature we have reviewed are described in decibel scale (dB) which is a well known log transform power scale used in engineering and biomedical research. Therefore, we think the log transform of the band powers is a reasonable methodology but are also reassured by the double check outlined above.

Moreover, during the correlation assessment between cortical beta and delta, the absence of power spectrum normalization adds to the inconsistency observed in the methodology. This non-uniformity might significantly influence the obtained results, warranting a clear justification for its presence in the analysis.

This part of the original manuscript could have been clearer. As both bandpowers were derived from the same source and signal, the risk of adding superficial negative correlation to their

interaction through normalization, (as described in detail above), was potentially more severe in this case and therefore we excluded normalization in the original manuscript. To illustrate this, we are presenting the same results with normalization:

Cortical delta and beta showed a very strong negative correlation in all PD participants (Panel E) during N2/N3 NREM (ON stimulation). LME analysis showed negative fixed effect of cortical beta power on cortical delta power ($\beta = -0.66$, 95% CI: [-0.7, -0.6], p-value = $1.3e-205$; n=241,643) during N2/N3 NREM sleep with a fixed group effect of PD/Dystonia state ($\beta = -0.04$, 95% CI: [-0.07, -0.014], p-value = 0.002).

Comparing these outcomes with our reported results and plots we see that the overall outcomes remains the same but here become artificially stronger with the addition of a normalization scheme (cortical delta-beta interaction LME: p-value= $1.3e-205$ for with normalization; p-value= $3.4e-5$ for without any normalization). Therefore to be conservative in our reporting, we elected not to normalize the signal for investigating cortical delta-beta interaction. We have now clarified this in our Methods section:

In order to avoid any risk of introduction of a potential spurious negative interaction in our single (cortical) site analysis we removed the normalization step for investigating cortical-cortical delta-beta interactions.

Reference:

1. Arnal, P. J. et al. The Dreem Headband compared to polysomnography for electroencephalographic signal acquisition and sleep staging. *Sleep* **43**, zsaa097 (2020).
2. Hammer, L. H., Kochanski, R. B., Starr, P. A. & Little, S. Artifact Characterization and a Multipurpose Template-Based Offline Removal Solution for a Sensing-Enabled Deep Brain Stimulation Device. *Stereotact. Funct. Neurosurg.* **100**, 168–183 (2022).
3. Neumann, W.-J. et al. The sensitivity of ECG contamination to surgical implantation site in brain computer interfaces. *Brain Stimul.* **14**, 1301–1306 (09/2021).
4. Mizrahi-Kliger, A. D., Kaplan, A., Israel, Z., Deffains, M. & Bergman, H. Basal ganglia beta oscillations during sleep underlie Parkinsonian insomnia *Supp. Proc. Natl. Acad. Sci. U. S. A.* **117**, 17359–17368 (2020).

REVIEWERS' COMMENTS

Reviewer #1 (Remarks to the Author):

The authors have thoroughly addressed all my comments. The paper has significantly improved through the revision. I particularly congratulate the authors for adding the board certified sleep expert which got even more convincing results. The notion that rises in beta activity prior to awakening is very important for ongoing aDBS trials.

Reviewer #2 (Remarks to the Author):

The authors adequately address the majority of my concerns, and in my opinion, the paper has improved substantially. However, the primary concern regarding the translative value of the findings for future adaptive Deep Brain Stimulation remains pending. Especially considering the broad readership of Nature Communications, a general translation to clinical practice is crucial. Given the limited availability of RC+S devices and the simultaneous implantation of STN/GPi and ECoG, it seems unlikely that this setup will become a clinical standard in the near future. Therefore, a transfer-learning of the individual Machine Learning (ML) model should be validated in an out-of-sample cohort with standard-of-care (sensing) DBS devices. If this translational aspect cannot be fulfilled, a more specialized journal within the Nature group may provide a better-targeted audience.

Reviewer Responses - Multi-night cortico-basal recordings reveal mechanisms of NREM slow-wave suppression and spontaneous awakenings in Parkinson's disease

We are grateful to the reviewers for their further comments that we have used to improve the manuscript substantially. We provide detailed responses to each suggestion below as well as key changes to the text.

- Reviewer comments in plain text
- *Author responses in italic*
- **Changes in the manuscript in bold**

Reviewer #1 (Remarks to the Author):

The authors have thoroughly addressed all my comments. The paper has significantly improved through the revision. I particularly congratulate the authors for adding the board certified sleep expert which got even more convincing results. The notion that rises in beta activity prior to awakening is very important for ongoing aDBS trials.

We thank the reviewer for acknowledging our work and highlighting the impact of the outcomes of this study in future research. We appreciate the thorough review of our manuscript and are pleased to know that our efforts in addressing the reviewer's comments have led to a significant improvement in the paper.

Reviewer #2 (Remarks to the Author):

The authors adequately address the majority of my concerns, and in my opinion, the paper has improved substantially.

We thank the reviewer for the thoughtful review. We are glad to hear that we have successfully addressed the majority of the concerns, and we appreciate the acknowledgment of the substantial improvement in the manuscript. The insights provided by the reviewer have been invaluable, and we are grateful for the opportunity to enhance the quality of our work based on the constructive feedback.

However, the primary concern regarding the translative value of the findings for future adaptive Deep Brain Stimulation remains pending. Especially considering the broad readership of Nature Communications, a general translation to clinical practice is crucial. Given the limited availability of RC+S devices and the simultaneous implantation of STN/GPi and ECoG, it seems unlikely that this setup will become a clinical standard in the near future. Therefore, a transfer-learning of the individual Machine Learning (ML) model should be validated in an out-of-sample cohort with standard-of-care (sensing) DBS devices. If this translational aspect cannot be fulfilled, a more specialized journal within the Nature group may provide a better-targeted audience.

We hear the reviewer's concern about the translational value of the findings in our study. However, the primary focus of our study was to investigate the mechanism of sleep dysfunction in Parkinson's disease (PD), in particular changes and interactions between subcortical beta and cortical slow-wave delta with and without Deep Brain Stimulation (DBS). Our findings indicated an inverse relationship between cortical delta and subcortical delta, for the first time in humans with PD and further - we linked this to spontaneous awakenings. While the reviewer is correct about the current limited clinical practice of simultaneous implantation of STN/GPi and ECoG our

core finding here was of a pathological rise in subcortical beta oscillations prior to spontaneous awakenings. This points to immediate potential translational value with current sensing-enabled neurostimulator systems which do not have cortical ECOG, but could be parameterized off subcortical beta during sleep. As highlighted by reviewer 1, such findings will play a crucial role in future aDBS trials as these data provide insight into the effect of DBS in sleep neurophysiology and open the possibility of providing early stimulation before awakening by tracking subcortical beta.

Notably, in this study, we also investigated the effects of DBS on cortical and subcortical neural activities during sleep (elevation in cortical delta and a decrease in alpha and low-beta) providing a deeper understanding of the role of how DBS might positively impact sleep. Overall, we think that while the technical setup of this study (i.e., simultaneous implantation of electrodes in cortical and subcortical areas) may not be the clinical standard at the current time, as indicated by the reviewer, the neurophysiological findings reported in this study can provide crucial foundations for a better understanding of the mechanism of sleep dysfunction in PD as well as supporting the design of sleep-specific aDBS strategies with currently available DBS neurostimulators. In summary - although it required research-grade DBS devices to understand the full network-level mechanism of sleep dysfunction in PD (subcortical beta → cortical delta). Cortical electrocorticography would not be required to design an adaptive DBS algorithm that can target pathological beta oscillations, as this biomarker is subcortical and could be targeted with current sensing-enabled DBS devices.

We had previously partly spoken about these points in the Discussion. Furthermore, we have now added the following to the Discussion as limitations:

The setup of this study, albeit naturalistic (at-home multi-night recordings), did employ investigational DBS hardware including the neurostimulator (Summit RC+S) and chronic cortical electrocorticography that are not widely available. This was necessary to elucidate the network-level mechanisms of sleep disruption in PD. However, the identified biomarker, subcortical beta, could be targeted using currently available devices and electrodes.

Also with reference to the reviewer's concerns and in regard to our ML models, we have provided region-specific ML models separately for subcortex and cortex avoiding across-region ML models in our work (that would be dependent on specialized hardware such as our setup). Indeed the subcortical models presented could also be run using currently available DBS systems. As such we believe our ML models are highly valuable and relevant to standard DBS hardware setups currently available. Finally, as we discussed previously, we built individual, patient-specific ML models, rather than a general model. As such our ML results were patient-specific which would not have been affected by adding more participants as suggested. We have added the following to the Discussion as limitations:

Finally, our ML models in this study were limited to straightforward binary classification (N2/N3 NREM vs. wakefulness) and were region-specific (trained separately for cortical and subcortical data) with a view toward the constraints of current and emerging sensing-enabled DBS devices. While we could not conduct out-of-sample tests due to the limited availability of data from similar setups, we utilized multiple cross-validation schemes for validating the performance of the ML models.